# The genetic architecture of NAFLD among inbred strains of mice

Simon T Hui[1]*, Brian W Parks[1], Elin Org[1], Frode Norheim[2], Nam Che[1], Calvin Pan[1], Lawrence W Castellani[1], Sarada Charugundla[1], Darwin L Dirks[1], Nikolaos Psychogios[3], Isaac Neuhaus[4], Robert E Gerszten[3,5], Todd Kirchgessner[6], Peter S Gargalovic[4], Aldons J Lusis[1]*

[1]Department of Medicine/Division of Cardiology, David Geffen School of Medicine, University of California, Los Angeles, Los Angeles, United States; [2]Department of Nutrition, Institute of Basic Medical Sciences, Faculty of Medicine, University of Oslo, Oslo, Norway; [3]Cardiovascular Research Center, Massachusetts General Hospital, Harvard Medical School, Boston, United States; [4]Department of Computational Genomics, Bristol-Myers Squibb, Princeton, United States; [5]Cardiology Division, Massachusetts General Hospital, Harvard Medical School, Boston, United States; [6]Department of Cardiovascular Drug Discovery, Bristol-Myers Squibb, Princeton, United States

**Abstract** To identify genetic and environmental factors contributing to the pathogenesis of non-alcoholic fatty liver disease, we examined liver steatosis and related clinical and molecular traits in more than 100 unique inbred mouse strains, which were fed a diet rich in fat and carbohydrates. A >30-fold variation in hepatic TG accumulation was observed among the strains. Genome-wide association studies revealed three loci associated with hepatic TG accumulation. Utilizing transcriptomic data from the liver and adipose tissue, we identified several high-confidence candidate genes for hepatic steatosis, including *Gde1*, a glycerophosphodiester phosphodiesterase not previously implicated in triglyceride metabolism. We confirmed the role of *Gde1* by in vivo hepatic over-expression and shRNA knockdown studies. We hypothesize that *Gde1* expression increases TG production by contributing to the production of glycerol-3-phosphate. Our multi-level data, including transcript levels, metabolite levels, and gut microbiota composition, provide a framework for understanding genetic and environmental interactions underlying hepatic steatosis.

*For correspondence: sthui@mednet.ucla.edu (STH); JLusis@mednet.ucla.edu (AJL)

**Competing interests:** The authors declare that no competing interests exist.

## Introduction

Non-alcoholic fatty liver disease (NAFLD) encompasses a wide spectrum of liver abnormalities, ranging from benign hepatocellular accumulation of lipids (steatosis), through non-alcoholic steatohepatitis (NASH), to fibrosis and cirrhosis in the absence of excessive consumption of alcohol and hepatitis viral infection. Advanced NAFLD can eventually progress to end-stage liver disease with increased risk of hepatocellular carcinoma (HCC) (*Kopec and Burns, 2011*). Population studies have shown that NAFLD is strongly associated with obesity, diabetes, and dyslipidemia (*Marchesini et al., 2003*). As such, NAFLD can be viewed as the hepatic manifestation of the metabolic syndrome. With the increasing prevalence of obesity, diabetes, and metabolic syndrome, it is not surprising that NAFLD is rapidly becoming the most common form of chronic liver disease worldwide (*Ratziu et al., 2010*). It is estimated that 20–30% of the population in Western developed countries is affected (*Vernon et al., 2011*). Despite the high prevalence of this disease, its natural history and etiology is poorly understood. Although simple steatosis appears to be benign and non-progressive in the

**eLife digest** Non-alcoholic fatty liver disease is a major health problem worldwide and is caused by an abnormal build-up of fat molecules in liver cells that disrupts how the cells work. Although many people with the disease show only mild or no symptoms, if the disease progresses the consequences—such as organ damage and an increased risk of liver cancer—can be severe.

Although non-alcoholic fatty liver disease has been linked with obesity and diabetes, how it develops is poorly understood. The most widely supported explanation suggests that the disease begins with an imbalance in the process that normally maintains the correct amount of fat molecules called triglycerides inside cells. As a result, triglycerides accumulate in the liver cells in a process known as steatosis, which is then thought to make the liver vulnerable to further problems. However, this theory has been questioned by genetic experiments that suggest triglyceride build-up actually protects cells from other kinds of damage.

Hui et al. studied mice that had been fed a diet that was high in fat and sugar. The extent of liver steatosis varied considerably between the mice, with some mice accumulating 30 times more triglyceride in their liver than others. The underlying variation in the genes of the mice was then examined to investigate whether this can explain the differences in liver condition. This revealed at least three DNA stretches that appear to be linked to triglyceride accumulation in the liver, including several genes that appear to be active during steatosis. One of these genes, known as Gde1, had not previously been shown to have a role in controlling how cells make and use triglycerides.

To confirm the role of Gde1, Hui et al. artificially turned the gene on in some mice and prevented it from turning on in others. Turning on Gde1 significantly increased the amount of triglyceride in the liver and keeping it turned off decreased triglyceride levels. Hui et al. suggest that this is because Gde1 helps to make a precursor molecule that is needed to build triglycerides. Certain gut bacteria also appear to be linked to steatosis.

This study used a population-based approach in mice to examine genetic factors in the development of fatty liver disease. The challenge now is to find out how the genes work and to understand their interactions with each other and with the environment.

---

majority of individuals affected with NAFLD (*Teli et al., 1995*), excessive fat accumulation in the liver is associated with organ pathology, including NASH and cirrhosis. NASH frequently progresses to fibrosis, cirrhosis, liver failure, and HCC, resulting in poor long-term prognosis.

The factors determining the progressive phenotype of this complex disease remain largely unknown, although it is clear that subtle genetic variations and environmental factors (such as diet and lifestyle) play a role in determining the disease phenotype and progression (*Day, 2002*; *Anstee et al., 2011*). According to the prevailing 'two-hit hypothesis' model (*Day and James, 1998*), it is believed that the first insult involves lipid accumulation in the hepatocytes, due to an imbalance in triglyceride homeostasis. Subsequently, steatosis increases the vulnerability of the liver to a 'second hit', which promotes liver injury, oxidative stress, inflammation, and fibrosis. However, silencing of diacylglycerol acyltransferase 2 (DGAT2) by an antisense oligo improved steatosis but worsened liver injury and fibrosis in mice (*Yamaguchi et al., 2007*). DGAT2-knockdown mice exhibit increased fatty acid (FA) oxidation through CYP2E1, leading to increased oxidative stress, inflammation, and tissue damage. These findings lead to the proposal that steatosis may be an adaptive response to protect the liver from lipotoxicity through partitioning toxic lipids into stable intracellular triglyceride stores. Hence, despite the fact that accumulation of TG was alleviated in DGAT2-knockdown liver, blocking this protective mechanism exacerbated lipotoxicity and led to more severe liver injury. These data suggest that TG accumulation per se is not the 'first hit' but the underlying inability to compensate for increased FA flux, which makes the liver prone to subsequent oxidative damage. It is proposed that the 'second hit' could involve diverse parallel pro-inflammatory signals derived from multiple sources (*Tilg and& Moschen, 2010*).

So far, there exists no established therapy targeting NAFLD. Since most NAFLD patients suffer from obesity and insulin resistance, treatment options aim at weight reduction, control of dyslipidemia and improving insulin sensitivity through lifestyle changes and pharmacological agents, such as metformin, statins, fibrates, and thiazolidinediones (*Schreuder et al., 2008*). Understanding the

underlying genetic factors contributing to NAFLD would not only fill the void of knowledge but would also facilitate the design of effective strategies in treating and preventing this important disease.

Considerable variations in the flux of TG and FAs occur in the liver in response to changing nutritional and hormonal status. Nevertheless, under normal physiological conditions, the liver does not accumulate considerable amounts of TG despite its active regulatory role in lipid trafficking. Processes that offset the balance of TG acquisition and disposition give rise to hepatic steatosis, which is the hallmark of NAFLD. These include increased FA uptake or de novo biosynthesis, reduced FA oxidation, and impaired lipoprotein production and secretion (*Farrell and Larter, 2006*). The FA moiety of hepatic TG is derived from three major sources: diet, de novo biosynthesis, and adipose tissue. In humans, it is estimated that 15% of hepatic TG comes directly from the diet, 26% from de novo lipogenesis, and 59% from adipose tissue in the form of non-esterified FAs (*Donnelly et al., 2005*). Together, the adipose tissue and liver play a major role in hepatic TG accumulation and supply >85% of FAs for the biosynthesis of TG in the liver.

Epidemiologic studies in human populations and animal studies indicate that genetics plays a substantial role in determining the susceptibility to the development of NAFLD (*Browning et al., 2004*; *Guerrero et al., 2009*; *Kahle et al., 2013*). The coexistence of NASH and cirrhosis clusters within families supports the existence of a common genetic link (*Struben et al., 2000*; *Willner et al., 2001*). The broad-sense heritability of hepatic steatosis has been estimated to be ~39% after adjustment for age, sex, race, and body mass index (*Schwimmer et al., 2009*). Although genetic predisposition to obesity and diabetes undoubtedly explains a fraction of the genetic component, additional independent genetic factors clearly contribute to the susceptibility to NAFLD (*Browning et al., 2004*). To date, only a small fraction of genes accounting for fatty liver disease have been identified and the molecular pathogenesis of NAFLD is poorly understood.

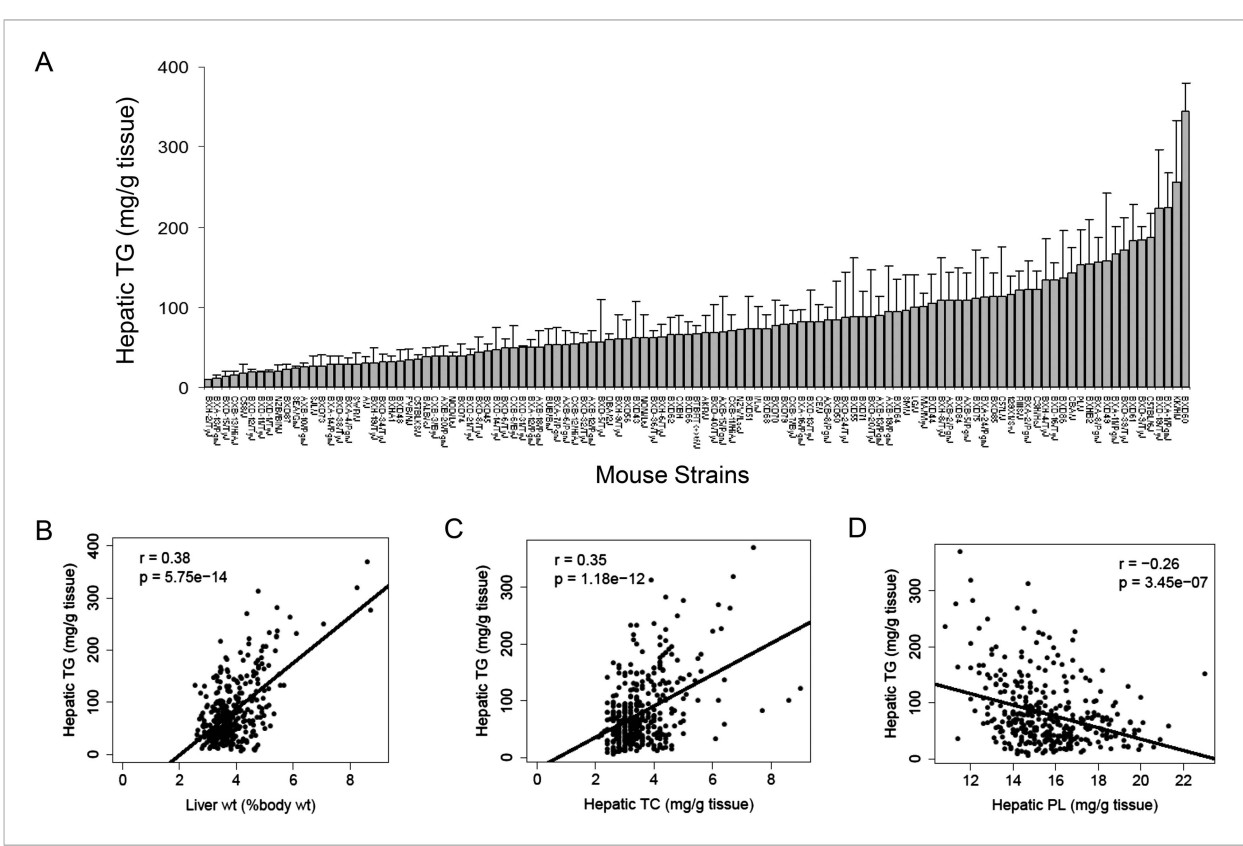

**Figure 1**. Effects of genetic background on hepatic TG accumulation. (**A**) Hepatic TG levels in male mice after 8 weeks of HF/HS feeding. Results are presented as mean + SD. (**B–D**) Correlation of hepatic TG with liver weight (**B**), hepatic total cholesterol (TC) (**C**), and hepatic phospholipid (**D**). r, biweight midcorrelation; p, p-value.

To gain insight into the etiology of NAFLD, we have employed a 'systems genetics' strategy to generate a comprehensive view of the genetic architecture of NAFLD in mice (*Civelek and Lusis, 2014*; *Mato et al., 2014*). We examined natural variations in the development of steatosis in a panel of inbred strains of mice fed a high-fat, high-sucrose (HF/HS) diet for 8 weeks. This diet has previously been shown to have a profound effect on obesity and insulin resistance depending upon the genetic background (*Parks et al., 2013*). We observed a wide range in phenotypes associated with hepatic steatosis following feeding the HF/HS diet. To further understand the molecular basis of steatosis, we employed a multi-omic approach to characterize the steatotic phenotype. Using global expression analysis of liver and adipose transcriptomes, we identified pathways enriched in strains susceptible to steatosis. We also showed that the susceptibility to steatosis is highly dependent on genetic background and was able to identify genetic loci contributing to steatosis. A causal gene at one of the loci was identified as *Gde1*. Furthermore, we identified metabolites and gut microbes, which are associated with steatosis. These multi-dimensional data provide a valuable resource for understanding the genetic–environmental interactions in the disorder.

## Results

### Large genetic variation in hepatic TG accumulation in mice fed with HF/HS diet

NAFLD is often referred to as the hepatic manifestation of metabolic syndrome, as it is associated with obesity, dyslipidemia, and insulin resistance (*Lazo and& Clark, 2008*). The development of NAFLD is strongly influenced by both dietary and genetic factors. We previously showed that the increase in

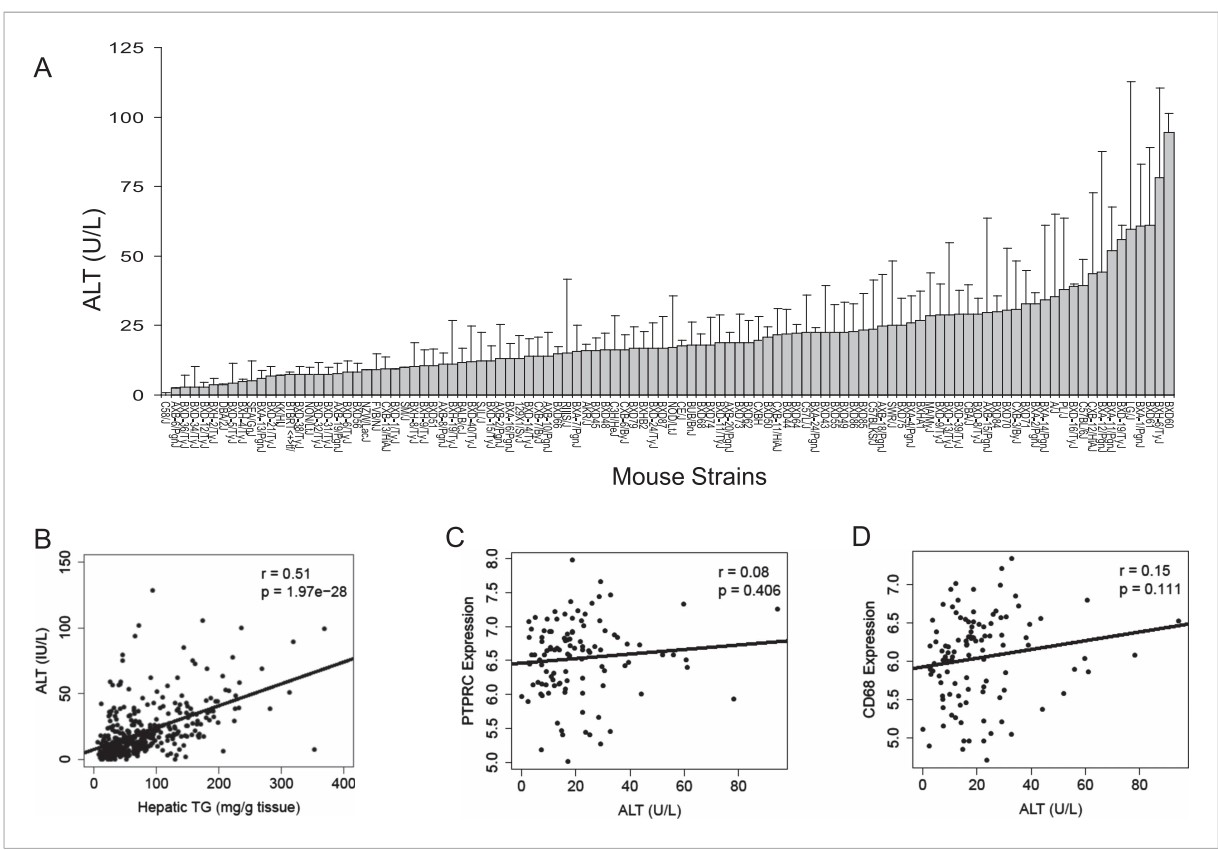

**Figure 2**. Plasma ALT activities and immune cell marker expression among inbred and recombinant inbred strains of mice. (**A**) Plasma alanine aminotransferase (ALT) activities in male mice after 8 weeks of HF/HS feeding. Results are presented as mean + SD. (**B–D**) Correlation of plasma ALT activities with hepatic triglyceride (**B**), hepatic *Ptprc* (*Cd45r*) expression (**C**) and hepatic. *Cd68* expression. r, biweight midcorrelation; p, p-value.

body weight and fat accumulation in response to a HF/HS diet in mice is highly dependent on the genetic background of individual strains (*Parks et al., 2013*). To study the gene-by-diet effects on hepatic steatosis, a panel of 8-week-old male HMDP mice were fed a HF/HS diet for 8 weeks to induce obesity and steatosis (*Parks et al., 2013*). Lipids were extracted and quantified in 478 individual livers from 113 strains of male mice. A wide spectrum of hepatic TG content was observed among the strains with more than 30-fold difference between the high and low responders (*Figure 1A* and *Supplementary file 1*). Hepatic TG content was significantly correlated with the liver weight (r = 0.38, p = 5.75 × 10$^{-14}$, *Figure 1B*). Contrary to the large variations in TG among strains, less than threefold difference in cholesterol and phospholipid levels was observed among the strains (*Supplementary file 1*). Modest correlations between these lipids with hepatic TG were observed. The TG content in the liver was positively correlated with hepatic total cholesterol (TC) content (r = 0.35, p = 1.18 × 10$^{-12}$, *Figure 1C*) but negatively correlated with the levels of phospholipids (r = −0.26, p = 3.45 × 10$^{-7}$, *Figure 1D*). These data suggest that increased neutral lipids content, in particular TG, contributes significantly to the enlarged livers.

To assess liver damage, we measured alanine aminotransferase (ALT) enzyme activity in the plasma (*Figure 2A*). Similar to hepatic TG content, we also observed a large variation in ALT activities among the strains. Plasma ALT activities showed a positive correlation with hepatic TG levels, implicating a role of increased hepatic TG levels in liver damage (*Figure 2B*). ALT is an established biochemical and clinical marker for liver damage as this enzyme is released into the circulation when the integrity of the cell membrane of hepatocytes is compromised. We performed histologic examination of livers from a subset of strains including some with high TG and did not observe evidence of significant pathology other than lipid accumulation (data not shown). Furthermore, mRNA levels of markers for B-lymphocytes (CD45/PTPPRC) and macrophages (CD68) were not increased in steatotic liver samples (*Figure 2C,D*). Similarly, markers for other immune cells, such as T cells (*Cd28, Csf2, Cd4, Ccr5, Gata3 Cxcr4*), B cells (*Pax5, Cd70, Cd79b*), and leukocytes (*Cd33, Cd52, Cd53, Cd44, Prg2*) were also not increased (data not shown), suggesting the absence of significant immune cell infiltration and steatohepatitis in these mice after 8 weeks on the HF/HS diet.

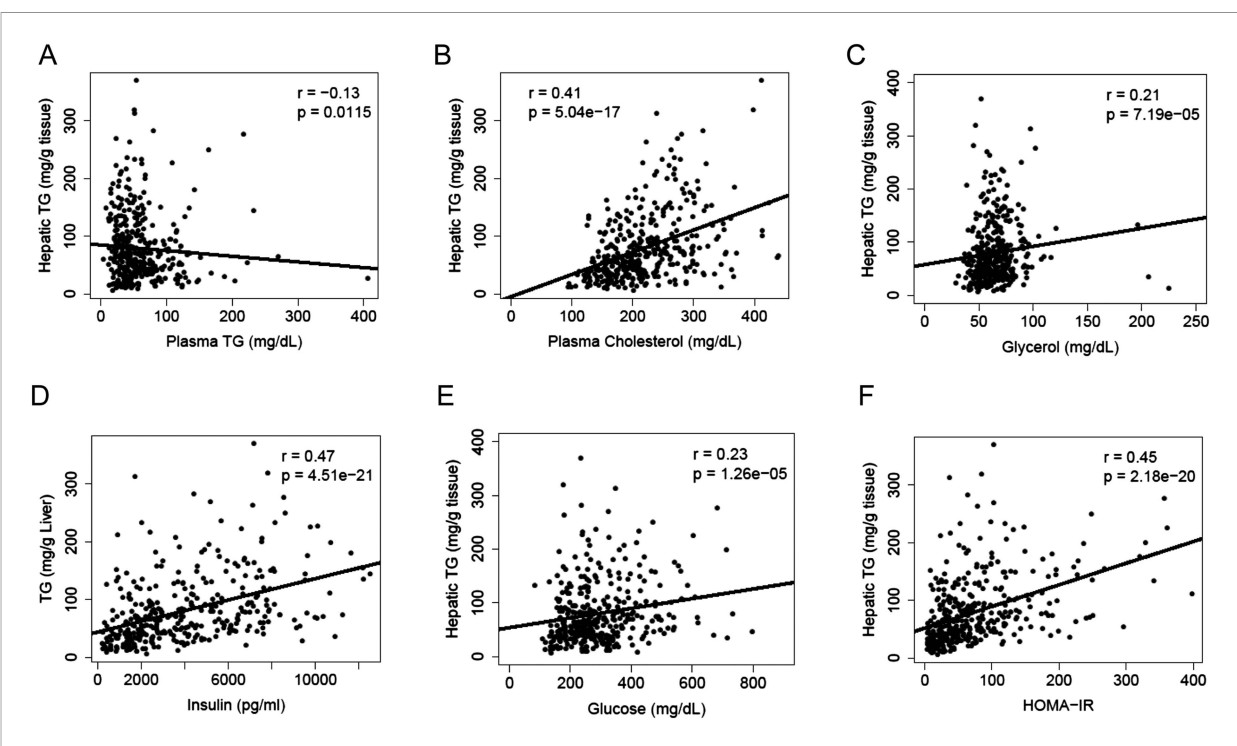

**Figure 3**. Correlation of hepatic TG content with plasma metabolites and HOMA-IR. (**A–F**) Correlation of hepatic TG with plasma TG (**A**), plasma cholesterol (**B**), plasma glycerol (**C**), plasma insulin (**D**), plasma glucose (**E**), and HOMA-IR (**F**). r, biweight midcorrelation; p, p-value.

## Hepatic TG accumulation is highly correlated with plasma cholesterol, insulin resistance, and adiposity

A substantial amount of hepatic TG is derived from FAs of extrahepatic sources, in particular, the white adipose tissue. We measured lipid and metabolites in the plasma and compared them to the hepatic TG content. Hepatic TG content was poorly correlated with plasma TG levels (r = −0.13, p = 0.0115, *Figure 3A*), whereas it was positively correlated with plasma cholesterol levels (r = 0.41, p = $5.04 \times 10^{-17}$, *Figure 3B*). The correlation between hepatic TG levels and plasma free FAs (FFAs) levels was not significant (r = 0.04, p = 0.44); however, hepatic TG levels were correlated with plasma glycerol levels (r = 0.20, p = 0.0001, *Figure 3C*), suggesting a link between liver steatosis and lipolysis in the adipose tissue. NAFLD is often associated with dyslipidemia (*Diehl et al., 1988*) and insulin resistance (*Marchesini et al., 1999*) in humans. Similar to the findings in humans, there was a robust association between hepatic steatosis and plasma insulin (r = 0.47, p = $4.51 \times 10^{-21}$, *Figure 3D*), glucose (r = 0.23, p = $1.26 \times 10^{-5}$, *Figure 3E*) as well as insulin resistance (HOMA-IR) (r = 0.45, p = $2.18 \times 10^{-20}$, *Figure 3F*).

Increased adiposity has been linked to the incidence of NAFLD in humans. Consistent with this finding, there was a robust correlation between hepatic TG levels and adiposity (r = 0.59, p = $6.70 \times 10^{-35}$, *Figure 4A*). We dissected the individual fat depots and found that steatosis was associated with increased subcutaneous, gonadal and mesenteric fat mass but not with retroperitoneal fat content (*Figure 4B–E*). The aforementioned p-values were not adjusted for multiple testing since the correlation analyses were performed based on knowledge of potential association between NAFLD and those clinical traits (e.g., insulin resistance, plasma lipids, and adiposity). Nevertheless, the correlations remained significant after Bonferroni correction for all measured traits.

## Transcriptomic analysis of the liver and adipose tissue

As liver and adipose tissues are primarily responsible for modulating liver TG accumulation, we performed transcriptomic analysis on these two tissues to gain insights into the pathogenesis of steatosis. *Tables 1, 2* show the top 50 most correlated genes with hepatic TG levels in the liver and adipose, respectively. None of the genes is in close proximity (<1.5 Mb) to each other on the same

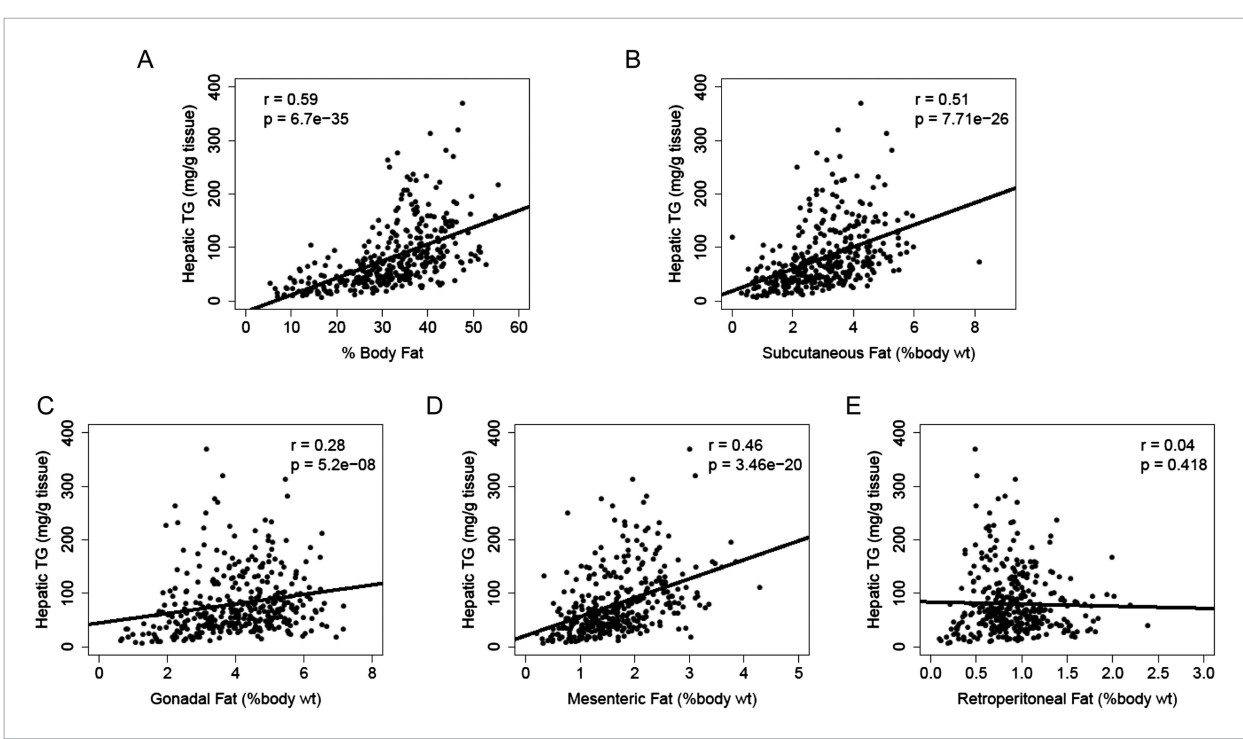

**Figure 4**. Correlation of hepatic TG content with adiposity and fat mass. (**A–E**) Correlation of hepatic TG with adiposity (**A**), subcutaneous fat (**B**), gonadal fat (**C**), mesenteric fat (**D**), and retroperitoneal fat (**E**). r, biweight midcorrelation; p, p-value.

**Table 1.** Top 50 liver genes correlated with hepatic TG levels

| Rank | Gene symbol | r | P |
|------|-------------|---|---|
| 1 | Cd36 | 0.695 | 1.85E-17 |
| 2 | Mrpl16 | 0.622 | 2.57E-13 |
| 3 | Enc1 | 0.611 | 8.12E-13 |
| 4 | 2010003K11Rik | 0.600 | 2.84E-12 |
| 5 | Tceal8 | 0.599 | 3.00E-12 |
| 6 | Cmpk1 | 0.598 | 3.32E-12 |
| 7 | Avpr1a | −0.595 | 4.45E-12 |
| 8 | Hmgcl | 0.590 | 7.60E-12 |
| 9 | Akap2 | 0.581 | 1.86E-11 |
| 10 | C1ra | −0.572 | 4.34E-11 |
| 11 | Reep3 | 0.562 | 1.16E-10 |
| 12 | Skp1a | 0.554 | 2.33E-10 |
| 13 | Esd | 0.551 | 3.11E-10 |
| 14 | Hadh | 0.550 | 3.48E-10 |
| 15 | Syap1 | 0.549 | 3.66E-10 |
| 16 | Ermp1 | 0.545 | 5.44E-10 |
| 17 | Ang | −0.544 | 5.53E-10 |
| 18 | Dak | 0.543 | 6.24E-10 |
| 19 | Matr3 | −0.541 | 7.12E-10 |
| 20 | Nudt9 | 0.538 | 9.35E-10 |
| 21 | Srsf5 | −0.538 | 9.89E-10 |
| 22 | Vps29 | 0.537 | 1.04E-09 |
| 23 | Ttc23 | 0.537 | 1.05E-09 |
| 24 | Entpd5 | 0.536 | 1.14E-09 |
| 25 | Chchd6 | 0.535 | 1.27E-09 |
| 26 | Plekha1 | 0.534 | 1.32E-09 |
| 27 | Mogat1 | 0.531 | 1.76E-09 |
| 28 | S100a10 | 0.528 | 2.11E-09 |
| 29 | Plin4 | 0.527 | 2.36E-09 |
| 30 | Anxa2 | 0.525 | 2.73E-09 |
| 31 | Srxn1 | 0.524 | 2.96E-09 |
| 32 | Cstb | 0.523 | 3.36E-09 |
| 33 | Cml1 | −0.522 | 3.70E-09 |
| 34 | Tpp1 | 0.521 | 4.00E-09 |
| 35 | Apoc2 | 0.518 | 5.06E-09 |
| 36 | F7 | −0.515 | 6.32E-09 |
| 37 | Wfdc2 | 0.514 | 6.72E-09 |
| 38 | Bche | 0.514 | 6.72E-09 |
| 39 | Mms19 | −0.514 | 6.74E-09 |
| 40 | Jun | 0.513 | 7.05E-09 |
| 41 | Lifr | −0.513 | 7.33E-09 |
| 42 | Gjb1 | −0.512 | 7.63E-09 |
| 43 | Fabp2 | 0.511 | 8.56E-09 |

*Table 1. Continued on next page*

chromosome, and thus, it is unlikely that some were correlated due to shared linkage disequilibrium (LD) blocks with biologically linked genes.

In the liver, *Cd36* had the highest correlation with TG levels (r = 0.70, p = $1.85 \times 10^{-17}$). CD36 is a multifunctional protein that enhances cellular FA uptake. Previous studies have shown that CD36-deficient mice are resistant to the induction of hepatic steatosis by alcohol and high-carbohydrate feeding (*Clugston et al., 2014*). Besides *Cd36*, many genes involved in lipid metabolism were also among the most correlated genes: *Hmgcl* (HMG-CoA lyase), *Hadh* (hydroxyacyl-CoA dehydrogenase), *Mogat1* (monoacylglycerol O-acyltransferase 1), *Plin4* (perlipin 4), *Apoc2* (apolipoprotein C-II), *Fabp2* (FA binding protein 2), *Slc16a7* (monocarboxylic acid transporters), and *Chpt1* (diacylglycerol cholinephosphotransferase). Interestingly, expression of the proto-oncogene c-Jun was positively correlated with hepatic TG content (r = 0.51, p = $7.05 \times 10^{-09}$). Enhanced hepatic c-Jun levels were observed in NAFLD patients, which correlated with inflammation and the degree of hepatic steatosis (*Dorn et al., 2014*). Increased c-Jun/AP-1 activation has been implicated in the progression of NAFLD (*Dorn et al., 2014*; *Hasenfuss et al., 2014*).

In the adipose, hepatic steatosis was significantly correlated with genes associated with adiposity and inflammation. *Prkcb* (protein kinase cβ), whose expression was significantly correlated with hepatic steatosis (r = 0.58, p = $1.62 \times 10^{-11}$), has been shown to be important in adipose tissue remodeling and FA metabolism (*Huang et al., 2012*). *Prkcb*-null mice are protected against diet-induced obesity and the development of hepatic steatosis and insulin resistance (*Huang et al., 2009*). *Cd53*, an adipose inflammatory marker, was also highly correlated with hepatic steatosis (r = 0.58, p = $2.06 \times 10^{-11}$). Previous microarray profiling showed that *Cd53* was significantly up-regulated in preadipocytes from obese human subjects (*Nair et al., 2005*).

Additionally, expression of many genes participating in immune response and inflammatory response was elevated in steatotic livers (*Bcl10*, *Il7r*, *Tlr7*, and *C1qb*). *Hcst* (hematopoietic cell signal transducer), previously shown to be a key hub gene in gene co-expression network that was significantly associated with serum TG levels (*Haas et al., 2012*) in humans, was also highly correlated with hepatic TG content (r = 0.57, p = $9.64 \times 10^{-11}$). The homeobox transcription factor Hoxa5 has previously been shown to be up-regulated after fat loss in human patients who

Table 1. Continued

| Rank | Gene symbol | r | P |
|------|-------------|------|----------|
| 44 | Morc4 | 0.510 | 9.22E-09 |
| 45 | Rnf11 | 0.510 | 9.33E-09 |
| 46 | Egfr | −0.509 | 1.03E-08 |
| 47 | Slc16a7 | 0.507 | 1.12E-08 |
| 48 | Gfm1 | 0.505 | 1.32E-08 |
| 49 | Chpt1 | 0.505 | 1.33E-08 |
| 50 | Rbbp4 | −0.505 | 1.33E-08 |

have undergone bariatric surgery (*Dankel et al., 2010*). In our study, *Hoxa5* expression was negatively correlated with hepatic TG content (r = −0.56, p = $1.37 \times 10^{-10}$). Human GWAS studies have shown that a missense mutation in LRRFIP1 was associated with adiposity and inflammation (*Plourde et al., 2013*). *Lrrfip1* expression was significantly associated with hepatic steatosis in our study (r = 0.56, p = $1.89 \times 10^{-10}$). *Adrb3* (β3-adrenergic receptor) activation induces white adipose remodeling and brown adipogensis (*Lee et al., 2012*). In our study, *Adrb3* expression was negatively correlated with hepatic TG levels (r = −0.56, p = $2.34 \times 10^{-10}$). Increased expression of *Lipa* (lysosomal acid lipase), which is involved in lysosomal lipophagy and TG/cholesterol ester catabolism, was found to be associated with hepatic steatosis (r = 0.56, p = $2.63 \times 10^{-10}$).

Enrichment analysis using the top 1000 hepatic genes correlated with hepatic TG levels (*Table 3*) showed a significant enrichment of mitochondrial genes (1.57 fold, adjusted p = $1.99 \times 10^{-5}$). Among the 127 mitochondrial genes, the majority (100 genes) were higher in steatotic livers, suggesting that altered mitochondria function is linked to the disease process of NAFLD. In addition, components of the extracellular matrix were enriched (2.91 fold, adjusted p = $5.89 \times 10^{-3}$) and were also predominantly (13 out of 17 genes) higher in steatotic livers (*Table 3*). Many of these genes are involved in wound healing and fibrosis, consistent with the observation of positive correlation between ALT and hepatic TG levels (*Figure 2B*). Complement and the coagulation cascade were specifically enriched (3.03 fold, adjusted p = $3.93 \times 10^{-3}$) and the genes were predominantly (15 out of 18 genes) lower in steatotic livers. In the adipose tissue, mitotic cell cycle, actin polymerization, cytoskeleton organization, immune response, response to wounding, leukocyte activation and positive regulation of cytokine production, lysosome pathway, and B cell receptor signaling were all enriched (*Table 4*). These findings suggest that inflammation in adipose tissue likely plays a role in the development of NAFLD. Enrichment analysis using the top 500 positively or negatively regulated genes separately did not reveal any additional enriched pathways.

The expression of several genes identified in human NAFLD GWAS also showed significant correlation with hepatic TG level (*Table 5*) (*Romeo et al., 2008*; *Chambers et al., 2011*; *Speliotes et al., 2011*; *Kozlitina et al., 2014*). In the liver, these included *Gckr* (r = 0.19, p = 0.044) and *Lyplal1* (r = 0.27, p = 0.003) and in adipose these included *Ncan* (r = 0.37, p = $6.1 \times 10^{-5}$), *Tm6sf2* (r = −0.23, p = 0.012), and *Trib1* (r = 0.24, p = 0.012).

## Identification of a genetic loci contributing to hepatic steatosis

To identify genetic loci associated with hepatic steatosis, we performed GWAS analysis on the hepatic TG contents with ~200,000 high-quality SNPs spaced throughout the genome. The genome-wide significant threshold was set at $1.31 \times 10^{-5}$, which corresponds to a 1% false discovery rate (FDR). We identified one genome-wide significant peak associated with hepatic TG content on chromosome 7 (*Figure 5A*). This peak does not overlap with any loci identified for obesity and insulin resistance in the same cohort of mice (*Parks et al., 2013*, *2015*). In addition, the peak locus on chromosome 7 still reached genome-wide significance when association mapping was conditioned using adiposity (% fat) or insulin resistance (HOMA-IR) as covariates (*Figure 6*), indicating that the effect of the chromosome 7 locus on hepatic steatosis is independent of obesity and diabetes. The peak SNP (rs32519715, p = $1.15 \times 10^{-6}$) falls in a LD block containing 17 genes (*Figure 5B* and *Table 6*). Three suggestive peaks, on chromosomes 3 (rs3708683; p = $5.47 \times 10^{-5}$), 9 (rs36804270; p = $1.30 \times 10^{-5}$), and 11 (rs13481015; p = $1.72 \times 10^{-5}$) passed the genome-wide significant threshold of 5% FDR (p = $1.34 \times 10^{-4}$). The peak on chromosome 9 also coincides with a genome-wide significant locus for insulin resistance (data not shown) and contains 13 genes within the LD block, whereas the peaks on chromosomes 3 and 11 contain 12 and 15 genes, respectively, within the associated LD blocks (*Table 7*).

**Table 2.** Top 50 adipose genes correlated with hepatic TG levels

| Rank | Gene symbol | r | p |
|------|-------------|------|----------|
| 1 | Nrbp2 | −0.624 | 2.42E-13 |
| 2 | Cp | −0.598 | 4.15E-12 |
| 3 | Hoxa7 | −0.591 | 8.42E-12 |
| 4 | Prkcb | 0.585 | 1.62E-11 |
| 5 | Cstb | 0.582 | 2.03E-11 |
| 6 | Cd53 | 0.582 | 2.06E-11 |
| 7 | Smap2 | 0.580 | 2.62E-11 |
| 8 | Sft2d1 | 0.578 | 3.04E-11 |
| 9 | Btk | 0.576 | 3.56E-11 |
| 10 | Was | 0.576 | 3.62E-11 |
| 11 | Il7r | 0.575 | 3.94E-11 |
| 12 | Tmem53 | −0.573 | 4.77E-11 |
| 13 | Rgs10 | 0.572 | 5.70E-11 |
| 14 | Srp19 | 0.571 | 6.27E-11 |
| 15 | Gpc3 | −0.570 | 6.53E-11 |
| 16 | Bcl10 | 0.570 | 6.72E-11 |
| 17 | Gpr65 | 0.570 | 6.82E-11 |
| 18 | Tlr7 | 0.569 | 7.18E-11 |
| 19 | Efhd2 | 0.569 | 7.26E-11 |
| 20 | Actr3 | 0.568 | 7.77E-11 |
| 21 | Cd72 | 0.568 | 8.02E-11 |
| 22 | Dera | 0.567 | 8.89E-11 |
| 23 | Pip4k2a | 0.566 | 9.63E-11 |
| 24 | Hcst | 0.566 | 9.64E-11 |
| 25 | Tyms | 0.566 | 9.78E-11 |
| 26 | Cenpv | −0.566 | 9.79E-11 |
| 27 | Plxnc1 | 0.565 | 1.05E-10 |
| 28 | Birc5 | 0.565 | 1.07E-10 |
| 29 | Ptpn18 | 0.563 | 1.27E-10 |
| 30 | Hoxa5 | −0.562 | 1.37E-10 |
| 31 | Fam105a | 0.562 | 1.43E-10 |
| 32 | Capza1 | 0.562 | 1.44E-10 |
| 33 | Nap1l3 | −0.561 | 1.50E-10 |
| 34 | Rgs18 | 0.561 | 1.51E-10 |
| 35 | Phtf2 | 0.560 | 1.59E-10 |
| 36 | Nckap1l | 0.560 | 1.65E-10 |
| 37 | Coro1c | 0.560 | 1.67E-10 |
| 38 | Coro1a | 0.559 | 1.78E-10 |
| 39 | Lrrfip1 | 0.559 | 1.89E-10 |
| 40 | C1qb | 0.557 | 2.13E-10 |
| 41 | Taok3 | 0.556 | 2.28E-10 |
| 42 | Ms4a6c | 0.556 | 2.29E-10 |
| 43 | Bco2 | −0.556 | 2.33E-10 |

*Table 2. Continued on next page*

## Integration of transcriptomic and association mapping data

The genetic variations underlying complex traits, such as steatosis, most often affect gene expression levels rather than structural (coding) aspects (*Wang et al., 2005*; *Hindorff et al., 2009*). Most of the large differences in gene expression are due to local differences (*Orozco et al., 2012*). Therefore, a useful approach to prioritizing candidate genes at a locus is to determine if the genes at the locus exhibit variation in expression that is controlled in *cis*. Such a variant is termed a *cis*-expression Quantitative Trait Locus (eQTL). We therefore identified significant *cis*-eQTL for liver and adipose for the genes in the chromosome 7 locus (*Table 8*). In addition, we asked whether the expression levels correlated with the clinical trait of interest (i.e., hepatic TG levels), since that would be consistent with a causal relationship. Among the candidate genes in the chromosome 7 locus, only three genes (*Coq7*, *Gde1*, and *Knop1*) have significant *cis*-eQTL and are also expressed in the liver (*Tables 6, 8*). These three genes also have significant *cis*-eQTL associations in adipose tissue (*Table 8*). The expression variation of these three candidate genes showed a continuous spectrum across the strains, indicating that the expression variations are not bimodal (data not shown). Hepatic TG levels correlated with *Gde1* expression in both the liver (r = 0.35, p = $1.5 \times 10^{-4}$) and adipose tissue (r = −0.21, p = $1.6 \times 10^{-3}$) (*Figure 7A,C*). Likewise, *Knop1* expression correlated with hepatic TG levels (r = 0.29, p = $2.5 \times 10^{-2}$, *Figure 7B*) but not in the adipose tissue (r = 0.14, p = 0.133). On the other hand, hepatic TG levels did not correlate with *Coq7* expression in the liver (r = 0.09, p = 0.368) or adipose tissue (r = −0.14, p = 0.142). *Gde1* (also known as *MIR16*) encodes glycerophosphodiester phosphodiesterase 1, a ubiquitously expressed enzyme involved in phospholipid metabolism, whereas *Knop1* encodes a lysine-rich nucleolar protein. Neither of these two genes has previously been identified in studies related to TG or lipid metabolism. We also examined coding variants for genes at the chromosome 7 locus using the PROVEAN prediction tool (*Choi and Chan, 2015*). A number of genes exhibited missense variants but these tended to be neutral and not likely to cause deleterious effects on protein stability and function (*Table 9*). Only the Q117R substitution in *Syt7* was predicted to be deleterious. *Syt7* (synaptotagmin VII) belongs to a protein family, which mediates $Ca^{2+}$-dependent vesicular

*Table 2. Continued*

| Rank | Gene symbol | r | p |
|------|-------------|------|----------|
| 44 | Adrb3 | −0.556 | 2.34E-10 |
| 45 | Arhgap9 | 0.556 | 2.37E-10 |
| 46 | Lrmp | 0.556 | 2.47E-10 |
| 47 | Fyb | 0.555 | 2.50E-10 |
| 48 | Lipa | 0.555 | 2.63E-10 |
| 49 | Cdt1 | 0.555 | 2.65E-10 |
| 50 | S100a4 | 0.554 | 2.80E-10 |

trafficking and exocytosis (*Moghadam and& Jackson, 2013*). Ablation of Syt7 has been shown to decrease insulin and glucagon secretion in pancreatic cells (*Gustavsson et al., 2008*, *2009*).

Based on *cis*-eQTL analyses, the chromosome 3 locus contains 3 strong candidate genes: *Smc4*, *Kpna4*, and *B3galnt1*. Expression of *Smc4* in adipose tissue significantly correlated with hepatic TG content (r = 0.51 and p = $9.8 \times 10^{-9}$). *Smc4* encodes structural maintenance of chromosomes 4-like 1, which is a core subunit of condensins I and II, large protein complexes involved in chromosome condensation and repair (*Onn et al., 2007*). Hepatic expression of *Kpna4* was correlated with liver TG content (r = 0.23 and p = 0.01). *Kpna4* encodes the subunit alpha-3 of importin, which is a cytoplasmic protein that recognizes nuclear localization signals of protein to be imported into the nucleus. Adipose expression of *B3galnt1,* encoding for the enzyme beta-1,3-galactosyltransferase 3, was correlated with steatosis (r = 0.30 and p = 0.001).

## Validation of Gde1 as a causal gene for hepatic steatosis at the chromosome 7 locus using adenoviral overexpression of Gde1 and shRNA knockdown

The finding of positive correlation between hepatic *Gde1* expression and steatosis suggests that increased expression of *Gde1* would promote hepatic TG accumulation. To directly assess the effect of *Gde1* on hepatic TG in vivo, *Gde1* was overexpressed in 8-week-old C57BL/6 mice (Ad-Gde1, $1 \times 10^9$ pfu per mouse, i.v.) by adenoviral transduction (*Figure 8A*). The control group (Ad-LacZ) received the same dose of adenovirus expressing LacZ. Mice were fed a HF/HS diet for 7 days after adenovirus injection and their hepatic lipids were measured. This regimen was chosen because gene expression by adenoviral transduction is only sustained for a short period of time (a few weeks or less). Preliminary studies in mice showed that HF/HS diet induced a threefold increase in hepatic TG accumulation in 1 week (data not shown). While there was no significant difference in body weight, the weight of livers from *Gde1*-overexpressing mice was 40% higher than that of the control (*Figure 8B*, p = $4.7 \times 10^{-5}$). In addition, plasma TG, TC, and FFA were all elevated in Ad-Gde1 mice (*Figure 8C*). MRI analysis showed that livers from *Gde1*-overexpressing mice contained significantly higher fat content (*Figure 8D*, p = 0.0002). Lipid analyses revealed that the increase in hepatic fat content was primarily due to increased accumulation of TG (*Figure 8E*, p = 0.014), as hepatic TC and phospholipid content were not significantly different from the control group (*Figure 8F,G*). Hepatic genes involved in TG biosynthesis (*Fasn*, *Dgat2*, and *Gpd1*) were down-regulated in mice overexpressing *Gde1* (*Figure 8H*).

Complementary to the overexpression studies, we also knocked down hepatic *Gde1* expression in vivo by adenoviral expression of shRNA. The virus dose and diet treatment were identical to that described above for the overexpression studies. Both *Gde1* protein and mRNA levels were ~75% decreased in the mice received adenoviral shRNA (*Figure 9A,B*). In contrast to the overexpression studies, knockdown of *Gde1* led to a significant decrease in hepatic TG content (*Figure 9C*, p = 0.011), whereas TC and phospholipid levels were not different from the control group (*Figure 9D,E*).

**Table 3**. Pathway-enrichment analysis of the top 1000 hepatic genes correlated with hepatic TG levels, assessed with the DAVID database, and presented as total genes meeting that criterion in each pathway (Count), along with Benjamini corrected p values (Adj. p)

| Category | Term | Count | Adj. p | Fold enrichment |
|----------|------|-------|--------|-----------------|
| GOTERM_CC_FAT | GO:0005739 ~ mitochondrion | 127 | 1.99E-05 | 1.57 |
| KEGG_PATHWAY | mmu04610:Complement and coagulation cascades | 18 | 3.93E-03 | 3.03 |
| GOTERM_CC_FAT | GO:0044420 ~ extracellular matrix part | 17 | 5.89E-03 | 2.91 |

**Table 4.** Pathway-enrichment analysis of the top 1000 adipose genes correlated with hepatic TG levels, assessed with the DAVID database, and presented as total genes meeting that criterion in each pathway (Count), along with Benjamini corrected p values (Adj. p)

| Category | Term | Count | Adj. p | Fold enrichment |
|---|---|---|---|---|
| GOTERM_BP_FAT | GO:0000087 ~ M phase of mitotic cell cycle | 36 | 9.69E-06 | 2.87 |
| KEGG_PATHWAY | mmu04142:Lysosome | 25 | 1.73E-03 | 2.50 |
| GOTERM_BP_FAT | GO:0008064 ~ regulation of actin polymerization or depolymerization | 19 | 5.97E-06 | 5.43 |
| GOTERM_BP_FAT | GO:0007010 ~ cytoskeleton organization | 39 | 1.89E-02 | 1.87 |
| GOTERM_BP_FAT | GO:0006955 ~ immune response | 50 | 1.80E-02 | 1.72 |
| GOTERM_BP_FAT | GO:0009611 ~ response to wounding | 42 | 2.64E-02 | 1.78 |
| GOTERM_BP_FAT | GO:0045321 ~ leukocyte activation | 34 | 7.64E-04 | 2.36 |
| KEGG_PATHWAY | mmu04662:B cell receptor signaling pathway | 20 | 8.62E-04 | 3.08 |
| GOTERM_BP_FAT | GO:0001819 ~ positive regulation of cytokine production | 13 | 1.85E-02 | 3.56 |

Similarly, expression of lipogenic genes (*Fasn*, *Dgat2*, and *Gpd1*) was increased in *Gde1*-knockdown livers (*Figure 9F*). These findings support a causal role for *Gde1* in hepatic steatosis under the chromosome 7 locus.

It is noteworthy that the direction of the effect of Gde1 expression with respect to hepatic TG levels in these studies is consistent with that observed in the HMDP in liver (i.e., a positive correlation).

## Plasma metabolites and hepatic steatosis

To identify metabolites associated with hepatic steatosis, we employed a metabolomic approach to measure 47 metabolites (amino acids, amines, and other polar compounds, *Supplementary file 2*) in the plasma of mice after 8 weeks of HF/HS feeding. Correlation analysis revealed a significant negative relationship between hepatic TG levels and plasma levels of arginine (r = −0.53, p = 9.85 × $10^{-12}$) and its degradative metabolite ornithine (r = −0.18, p = 0.027) (*Figure 10A,B*), whereas citrulline, another degradative metabolite of arginine, showed a positive correlation with hepatic TG content (r = 0.18, p = 0.034, *Figure 10C*). Hepatic TG levels were positively correlated with plasma levels of trimethylamine-N-oxide (TMANO, r = 0.18, p = 0.034, *Figure 10D*). Increased TMANO levels have previously been implicated in the susceptibility of strain 129S6 mice to diet-induced impaired glucose homeostasis and NAFLD (*Dumas et al., 2006*). TMANO is an oxidative product of trimethylamine (TMA), a metabolite of choline in animals. TMANO levels are regulated by both genetic and dietary factors and are strongly associated with atherosclerosis (*Bennett et al., 2013*). The oxidation of TMA is catalyzed by the hepatic flavin-containing monooxygenase (FMO) family of enzymes with FMO3 having the highest catalytic activity. No significant correlation between FMO3 expression and hepatic TG content was observed; however, among the 5 members in the FMO family, FMO5 expression exhibited a significant correlation with hepatic TG content (r = 0.46, p = 2.71 × $10^{-7}$). Hepatic TG content was also positively associated with plasma creatine (r = 0.23, p = 6 × $10^{-3}$, *Figure 10E*) and creatinine levels (r = 0.25, p = 6 × $10^{-3}$, *Figure 10F*).

**Table 5.** Correlation between human GWAS candidate gene expression in mouse liver and adipose tissue with hepatic TG level

| | Liver | | Adipose | |
|---|---|---|---|---|
| | r | p | r | p |
| *Pnpla3* | 0.07 | 0.423 | −0.04 | 0.645 |
| *Gckr* | 0.19 | 0.044* | 0.14 | 0.156 |
| *Ncan* | −0.10 | 0.316 | 0.37 | 6.1 × 10-5* |
| *Tm6sf2* | 0.15 | 0.123 | −0.23 | 0.012* |
| *Lyplal1* | 0.27 | 0.003* | −0.12 | 0.228 |
| *Trib1* | −0.10 | 0.313 | 0.24 | 0.012* |

*Denotes p < 0.05.

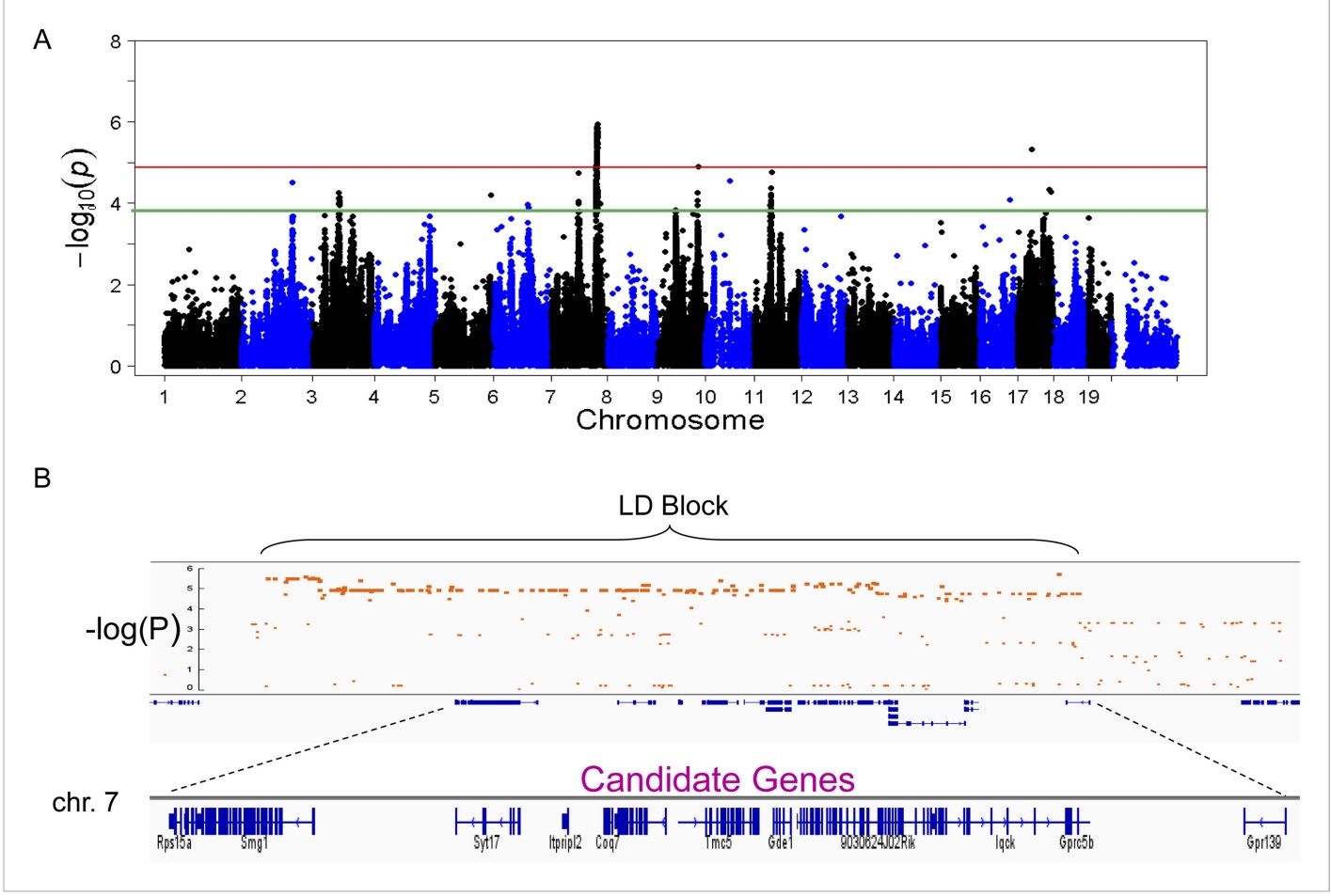

**Figure 5**. Association mapping of hepatic TG. (**A**) Manhattan plot showing the significance (−log of p) of all SNPs and hepatic TG. Genome-wide significance cut-off at 1% false discovery rate (FDR) is shown by the red line and cut-off at 5% FDR is shown in green. (**B**) Locus plot for genome-wide significant locus on chromosome 7 with approximate linkage disequilibrium block and candidate genes.

## Association of gut microbiota with hepatic lipid accumulation

Gut microbiota has been implicated in certain human inflammatory and metabolic diseases, including obesity (*Ley et al., 2006*; *Turnbaugh et al., 2009*), inflammatory bowel disease (*Lepage et al., 2011*), and type II diabetes (*Qin et al., 2012*). Accumulating evidence also suggests that alterations in gut microbiota composition could contribute to the susceptibility and progression of NAFLD (*Mouzaki et al., 2013*; *Zhu et al., 2013*). To understand the relationship between microbiota composition and hepatic lipid accumulation in mice fed a HF/HS diet, we employed deep sequencing of a conserved region of the bacterial 16S ribosomal RNA gene to determine the cecal microbiota composition in 237 male mice among 100 different strains. There was no significant correlation between hepatic TG levels and overall microbial diversity, as determined by Shannon diversity index (*Mills and& Wassel, 1980*). For the bacterial families and genera detected at significant levels, none were significantly associated with hepatic TG levels (5% FDR) (*Table 10*). However, associations between certain taxa were observed with other hepatic lipids (*Table 10*). *Coprococcus* and *Oscillospira* were positively associated with hepatic unesterified cholesterol levels, whereas negative associations were observed with *Blatuia*. *Blautia* and *Allobaculum* were negatively associated with TC. *Coprococcus* and *Oscillospira* were positively associated with phospholipids, whereas *Ruminococcus* showed negative association (*Table 10*).

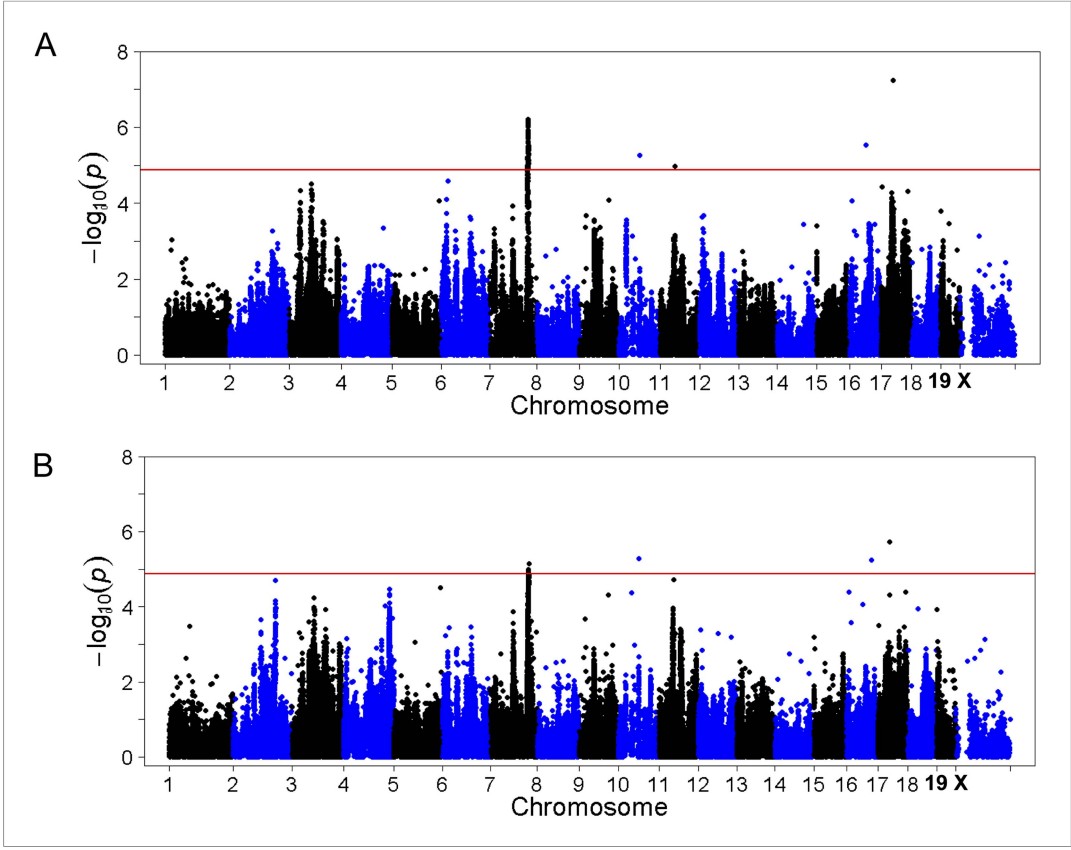

**Figure 6**. Association mapping of hepatic TG with adiposity or insulin resistance as covariates. Manhattan plot showing the significance $-\log_{10}$ (p value) of all SNPs and hepatic TG conditioned for percentage body fat (**A**) or HOMA-IR (**B**) as covariates. Genome-wide significance cut-off at 1% FDR is shown by the red line.

## Discussion

Along with the increased prevalence of obesity and diabetes, NAFLD has become the most common cause of chronic liver disease in a number of Western countries including the United States (*Ratziu et al., 2010*; *Vernon et al., 2011*). The etiology and pathogenesis of NAFLD are still poorly understood, reflecting the genetic and environmental heterogeneity of the disease. The molecular basis of the metabolic response to dietary fat and its role in NAFLD development remains to be elucidated. Previous studies using inbred mouse strains have shown that genetic background affects lipid accumulation in the liver (*Hill-Baskin et al., 2009*; *Millward et al., 2009*; *Shockley et al., 2009*; *Montgomery et al., 2013*). In this study, we employed over 100 strains of inbred and recombinant inbred mice and demonstrated vast genetic variation in hepatic TG accumulation in fed with a HF/HS diet. We examined the relationship between NAFLD and insulin resistance, obesity, plasma metabolites, plasma lipoproteins, and gut microbiota. We also carried out global gene expression array analyses on two key metabolically relevant tissues for NAFLD (liver and white adipose tissue) and identified molecular pathways enriched in NAFLD. Finally, we identified genetic loci contributing to NAFLD in common inbred strains of mice and validated the role of a novel gene, *Gde1*, at one of the loci. Our data substantially expand our knowledge of the genetic architecture of NAFLD and provide a rich resource for future biochemical and genetic studies.

While there is strong evidence that genetic factors contribute significantly to the onset and progression of the disease, only a few genes have been identified by human GWAS studies (*Browning et al., 2004*; *Romeo et al., 2008*; *Chalasani et al., 2010*; *Adams et al., 2013*; *Kozlitina et al., 2014*). Because genetic and environmental factors can be strictly controlled in mice, such models are valuable for dissecting the genetic and environmental contributions to complex disease traits. Nevertheless,

**Table 6**. Association p-values of candidate genes on chromosome 7 with the peak SNP for hepatic triglyceride levels

| Gene | Liver p-value | Liver expression | Adipose p-value | Adipose expression |
|---|---|---|---|---|
| Rps15a | NA | Yes | NA | Yes |
| Arl6ip1 | 0.011 | Yes | 0.294 | Yes |
| Smg1 | NA | No | NA | Yes |
| 4930583K01Rik | NA | No | NA | Yes |
| Syt7 | 0.495 | No | 0.748 | Yes |
| Itpripl2 | NA | Yes | NA | Yes |
| Coq7 | 7.18E-13 | Yes | 7.96E-14 | Yes |
| Tmc7 | NA | Yes | NA | Yes |
| Tmc5 | NA | No | NA | Yes |
| Gde1 | 3.21E-05 | Yes | 3.98E-08 | Yes |
| Ccp110 | NA | No | NA | No |
| 9030624J02Rik | NA | Yes | NA | Yes |
| Knop1 | 3.85E-07 | Yes | 0.043 | Yes |
| Iqck | NA | No | NA | No |
| Gprc5b | 0.002 | No | 0.661 | Yes |
| Gpr139 | NA | Yes | NA | No |
| Gp2 | 0.910 | No | 0.209 | No |

classical linkage analysis approaches to dissect NAFLD in mice have been relatively unsuccessful. While a number of loci have been identified by quantitative QTL analyses, a major problem has been an inability to carry out fine genetic mapping to identify the responsible genes (*Rangnekar et al., 2006*; *Kumazawa et al., 2007*; *Minkina et al., 2012*). In this study, we employed an innovative genome-wide association approach using >100 strains of inbred and recombinant inbred strains of

**Table 7**. Candidate genes under the chromosomes 3, 9, and 11 loci

| Chromosome 3 | Chromosome 9 | Chromosome 11 |
|---|---|---|
| 1110032F04Rik | Nphp3 | Il12b |
| Ift80 | Uba5 | Ublcp1 |
| Smc4 | Acad11 | Rnf145 |
| Trim59 | Dnajc3 | Ebf1 |
| Kpna4 | Acpp | Gm12159 |
| Gm1647 | Cpne4 | F630206G17Rik |
| Arl14 | Mrpl3 | Clint1 |
| Ppm1l | Nudt16 | Lsm11 |
| B3galnt1 | 1700080E11Rik | Thg1l |
| Nmd3 | Nek11 | Sox30 |
| Sptssb | Aste1 | Adam19 |
| Otol1 | Atp2c1 | Nipal4 |
|  | Pik3r4 | Cyfip2 |
|  |  | Itk |
|  |  | Fam71b |

**Table 8**. Significant cis-eQTL at chromosome 7 locus

| Gene | Gene start position | SNP ID | Position | Liver p-value |
|---|---|---|---|---|
| *Arl6ip1* | 118118891 | rs30668041 | 118266969 | 2.79E-24 |
| *Coq7* | 118509659 | rs32461510 | 118350009 | 5.01E-24 |
| *Gde1* | 118688545 | rs32511419 | 119070521 | 1.32E-06 |
| *Knop1* | 118842237 | rs32246745 | 119160823 | 1.18E-07 |
| **Gene** | **Gene start position** | **SNP ID** | **Position** | **Adipose p-value** |
| *Arl6ip1* | 118118891 | rs30668041 | 118266969 | 1.14E-36 |
| *Coq7* | 118509659 | rs32430851 | 117961092 | 1.65E-18 |
| *Gde1* | 118688545 | rs31516425 | 118372786 | 2.53E-12 |
| *Knop1* | 118842237 | rs32532370 | 119288974 | 3.47E-10 |
| *Gprc5b* | 118972040 | rs48647926 | 118918455 | 3.70E-09 |

eQTL, expression Quantitative Trait Locus.

mice to finely map genetic loci contributing to the development of steatosis. This approach circumvents the obstacles associated with human studies (namely, environmental heterogeneity) and mouse linkage analyses (namely, poor mapping resolution). Furthermore, by integrating transcriptomic information for the liver and adipose tissue, we identified two high-confidence candidate genes (*Gde1* and *Knop1*) for hepatic steatosis on chromosome 7. We pursued *Gde1* using experimental perturbation based on its known role in lipid metabolism.

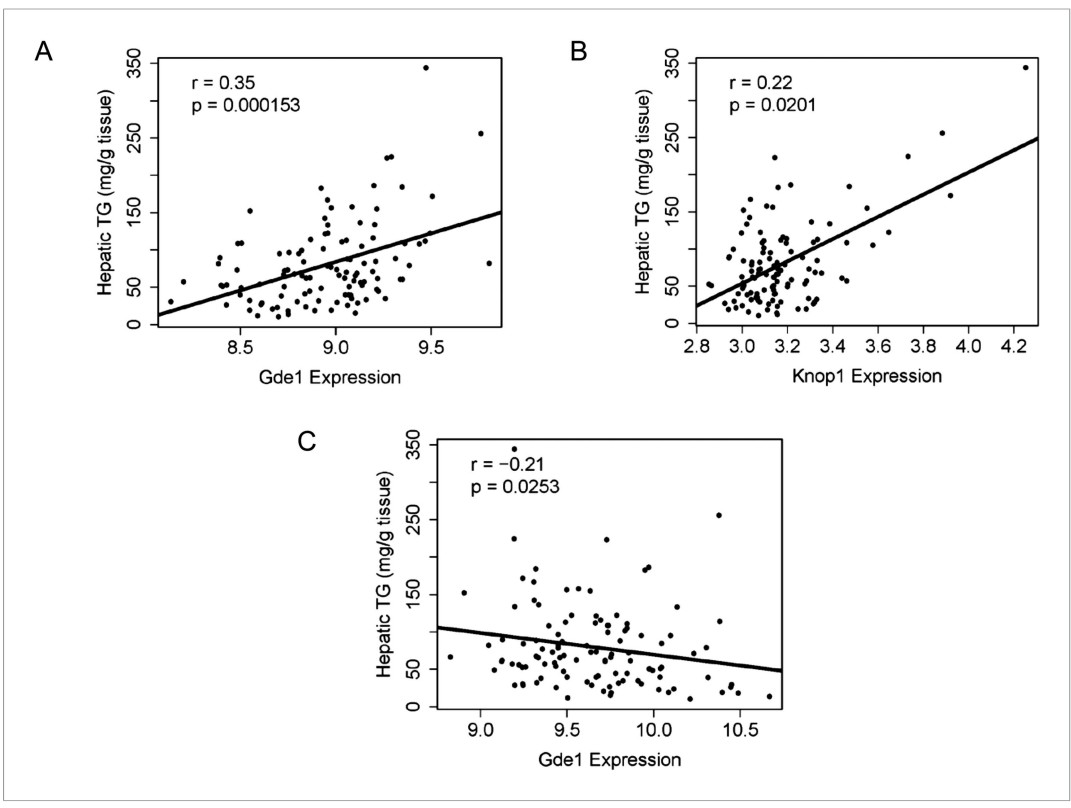

**Figure 7**. Correlation of candidate gene expression with hepatic TG content. (**A–B**) Correlation of hepatic TG with expression levels of *Gde1* (**A**) and *Knop1* (**B**) in the liver. (**C**) Correlation of hepatic TG with expression levels of *Gde1* in the white adipose tissue. r, biweight midcorrelation; p, p-value.

**Table 9**. Missense variants of candidate genes on chromosome 7

| Gene | Missense variants |
| --- | --- |
| Rps15a | None |
| Arl6ip1 | None |
| Smg1 | None |
| 4930583K01Rik | None |
| Syt7 | Q117R |
| Itpripl2 | R204H, V240L, S420G |
| Coq7 | A290G |
| Tmc7 | C73S, T358S |
| Tmc5 | M11V, Q42L, V82D, A119T, V139I, P179S, A243G, R258K, V448I, E737D |
| Gde1 | None |
| Ccp110 | R180K, I199V, N248S, A326V, A332T, P427T, S439P, S445P, F624S, G746S |
| 9030624J02Rik | S3A, D23G, V122A, G175A, T660M |
| Knop1 | I70M, V49A |
| Iqck | V49A, S148N |
| Gprc5b | None |
| Gpr139 | None |
| Gp2 | V79D, R483G |

The effect of missense mutation was assessed by PROVEAN software. Neutral amino acid substitutions, which do not affect protein stability and function are shown in blue whereas deleterious mutations are labeled in red.

Several lines of evidence indicate that *Gde1* is a causal gene for hepatic steatosis in the chromosome 7 locus. First, the presence of *cis*-eQTL suggests that *Gde1* expression is regulated locally, associating with the SNP haplotype. Second, hepatic expression of *Gde1* is significantly correlated with the levels of hepatic TG. Third, overexpression of *Gde1* promoted specifically TG accumulation in vivo, whereas hepatic cholesterol and phospholipid levels were unaffected. Forth, shRNA knockdown of *Gde1* reduced hepatic TG accumulation.

*Gde1* encodes glycerophosphodiester phosphodiesterase 1 (EC 3.1.4.46), a broadly expressed integral membrane glycoprotein, which catalyzes the degradation of glycerophosphoethanolamine and glycerophosphocholine (*Okazaki et al., 2010*; *Simon and& Cravatt, 2010*). Two forms of the protein (~37 kDa and 43 kDa) were detected in Western blots of endogenous and adenoviral overexpressed GDE1, which is likely be due to variable glycosylation as previously documented (*Zheng et al., 2000*). Its role in triglyceride metabolism has not been documented. Expression levels of TG biosynthetic pathways in the livers of *Gde1*-overexpressing mice were down-regulated, indicating that increased de novo lipogenesis due to increased lipogenic gene expression is unlikely the cause of TG accumulation. Increased expression of *Cd36* in *Gde1*-overexpressing livers raises the possibility that *Cd36* acts as a mediator of the effect of *Gde1* through its lipid transport activity. One of the products of the *Gde1* enzymatic reaction is glycerol-3-phosphate, which can be converted to phosphatidate by glycerol-3-phosphate acyltransferase (GPAT) and subsequently dephosphorylated by the phosphatidate phosphatase lipin to form diacylglycerol (DAG). DAG can then be acylated by DGAT to form TG. We hypothesize that *Gde1* affects the availability of glycerol-3-phosphate and modulates the flux of TG in the liver. The negative correlation of adipose *Gde1* expression and hepatic TG may be due to a different role of glycerol-3-phosphate in the adipose tissue. Increased glycerol-3-phosphate production (when *Gde1* expression is high) in the adipose tissue would promote FA re-esterification, leading to a decrease in FA release to the circulation. This reduction in FA supply to the liver may result in diminished TG synthesis. The opposite pattern of regulation of *Gde1* implicates tissue-specific regulatory elements. Mouse ENCODE data (*Yue et al., 2014*) showed that there are differences in DNA hypersensitive sites in the gene region, suggesting that

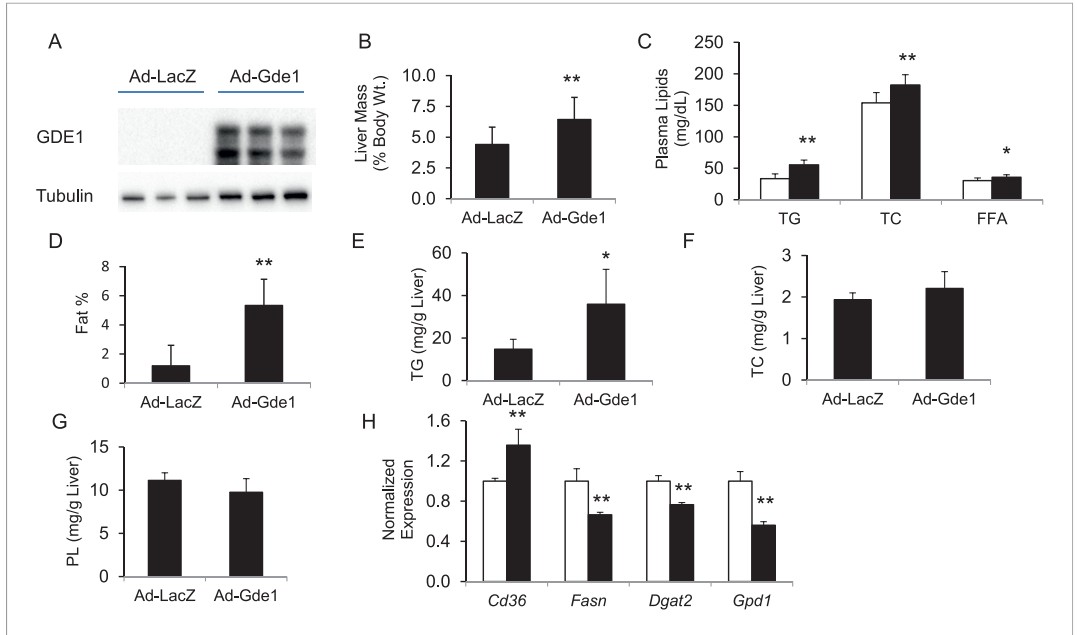

**Figure 8**. Effects of *Gde1* overexpression in mice by adenoviral transduction. C57BL/6 mice were injected with Ad-Gde1 ($1 \times 10^9$ pfu per mouse, *i.v.*) and fed with a HF/HS diet for 7 days. Control group received the same dose of Ad-LacZ. (**A**) Western-blot of liver homogenate using anti-GDE1 or anti-tubulin antibody. (**B**) Comparison of liver weight between *Gde1*-overexpressing mice and the control mice. (**C**) Differences in plasma triglyceride (TG), TC, and free fatty acids levels between *Gde1*-overexpressing mice (filled bars) and the control mice (empty bars). (**D**) Hepatic fat percentage in the two groups of mice was determined by MRI. (**E–G**) Liver lipids were extracted and quantified: triglyceride (TG), TC, and phospholipids (PL). (**H**) Expression of lipogenic genes was measured by qPCR and normalized to the level of the housekeeping gene *36B4*. Ad-LacZ (empty bars) and Ad-Gde1 (filled bars) Results are presented as mean + SD (n = 7–8) * denotes p < 0.05 and ** denotes p < 0.01.

there may be differences in DNA accessibility and transcription factor binding in the liver and adipose tissue.

The candidate genes under the chromosome 9 locus are likely to affect hepatic TG content via an insulin-dependent manner as the peak of association was diminished by co-mapping with obesity or insulin resistance. Insulin is a key hormone that drives lipogenesis and hepatic steatosis is often accompanied by hepatic insulin resistance. Our data showed that hepatic TG load has a robust association with plasma insulin levels and HOMA-IR (*Figure 6*). Impaired insulin signaling in the liver leads to the failure of insulin to suppress gluconeogenesis through the FoxO1 pathway, leading to hyperglycemia and ultimately diabetes (*Haas and Biddinger, 2009*; *Leavens and& Birnbaum, 2011*). Paradoxically, insulin-stimulated hepatic lipogenesis through SREBP-1c induction is not impaired in steatosis-associated insulin resistant livers (*Brown and Goldstein, 2008*; *Li et al., 2010*). This selective insulin resistance leads to increased production of lipids and steatosis. The chromosome 3 signal was not affected by conditioning on percentage body fat or HOMA-IR, suggesting that the causal gene(s) determining hepatic TG content at this locus are unlikely to be mediated by pathways involving body fat or insulin sensitivity.

Our observed enrichment of mitochondrial genes in steatotic livers suggests that disrupted mitochondrial bioenergetics may play a role in NAFLD pathobiology. This is in accordance with finding that chronic consumption of a HF diet-induced NAFLD with reduced mitochondrial oxidation and increased ROS production (*Mantena et al., 2009*). Obesity-induced steatosis has also been linked to decreased hepatic ATP synthesis (*Chavin et al., 1999*). These findings highlighted the importance of mitochondrial function in the pathogenesis of NAFLD. Variations in mitochondrial capacity and activity may contribute to the differences in susceptibility to HF/HS diet challenge. Strains with higher mitochondrial capacity would be more resistant to the development of NAFLD due to more efficient oxidation and disposal of surplus nutrients and lower production of ROS as a result of more efficient coupling between oxidation and phosphorylation. Determining the connection between the genetic

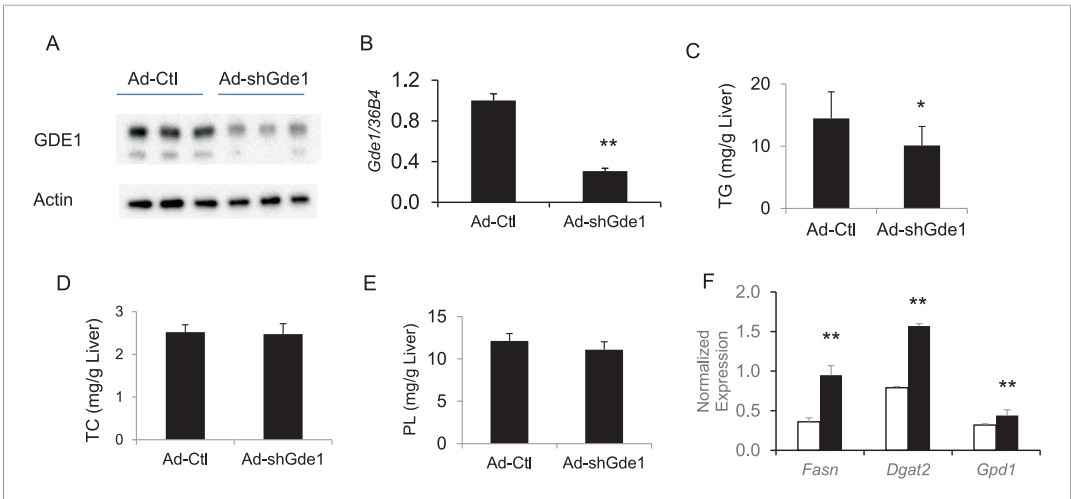

**Figure 9**. Effects of *Gde1* knockdown in mice by adenoviral transduction. C57BL/6 mice were injected with Ad-shGde1 ($1 \times 10^9$ pfu per mouse, *i.v.*) and fed with a HF/HS diet for 7 days. Control group received the same dose of Ad-Ctl. (**A**) Equal amounts of liver protein were loaded in each lane and Western-blotted using anti-GDE1 or anti-actin antibody. (**B**) Comparison of *Gde1* mRNA levels between *Gde1*-knockdown mice and the control mice. (**C–E**) Liver lipids were extracted and quantified: triglyceride (TG), TC, and phospholipids (PL). (**F**) Expression of lipogenic genes was measured by qPCR and normalized to the level of the housekeeping gene *36B4*. Results are presented as mean + SD (n = 11–12) * denotes p < 0.05 and ** denotes p < 0.01.

determinants of mitochondrial bioenergetics and hepatic TG accumulation could provide important insights into the etiology of NAFLD.

Metabolomics has been used to identify biomarkers for NAFLD (reviewed in *Dumas et al., 2014*). Our analysis showed that hepatic steatosis was associated with low-plasma levels of arginine. The inverse relationship between arginine levels and hepatic TG content may be due to increased arginine degradation in steatotic mice. Previous studies have shown that an arginine-deficient diet-induced hepatic steatosis in rats (*Milner and& Hassan, 1981*). The levels of arginase 1, which catabolizes L-arginine to urea and L-ornithine, were increased in steatotic livers from mice fed a high-fat diet (*Eccleston et al., 2011*). Exposure to HF diet led to decreased hepatic levels of activated endothelial nitric oxide synthase (eNOS), which converts L-arginine to nitric oxide (NO) and citrulline. In our study, gene expression of *Arg1* and *Nos3* did not show a significant correlation with hepatic TG content, suggesting the effect of HF diet on the activities of these two enzymes acts post-transcriptionally. Arginine plays an important role in maintaining the integrity of cell junctions and in the regulation of the epithelial barrier (*Marc Rhoads and& Wu, 2009*). It also modulates immune response and is essential for tissue healing (*Wu et al., 2009*). Low levels of arginine may lead to increased gut bacterial and endotoxin translocation, which promotes liver inflammation and NAFLD progression. We observed that TMANO was positively correlated (r = 0.18, p = 0.034) with hepatic steatosis; however, the correlation was not significant after correction for multiple testing. Previous results from studies of 129S6 mice, a model of diet-induced fatty liver disease, showed that TMANO was a marker for NAFLD (*Dumas et al., 2006*). Increased TMANO levels may indicate increased conversion of dietary choline into methylamines by microbiota, leading to reduced bioavailability of choline and mimicking the effect of a choline-deficient diet.

Our study also identified *Coprococcus* and *Oscillospira* as potentially important microbes in lipid homeostasis, as the abundance of these microbes showed significant association with levels of cholesterol and phospholipids. Studies in humans showed a large reduction in abundance of these microbes in obese and NASH subjects (*Zhu et al., 2013*). Little is known about the biochemical activities of these microbes; however, it has been shown that dietary composition could affect their abundance in humans (*Walker et al., 2011*). Studies of diet-induced obesity in mice suggest that abundance of *Oscillospira* may affect gut barrier function in the proximal colon (*Lam et al., 2012*). Further analysis of the genomic sequences to identify the specific strains involved could provide new insights into the role of these microbes in the pathogenesis of NAFLD.

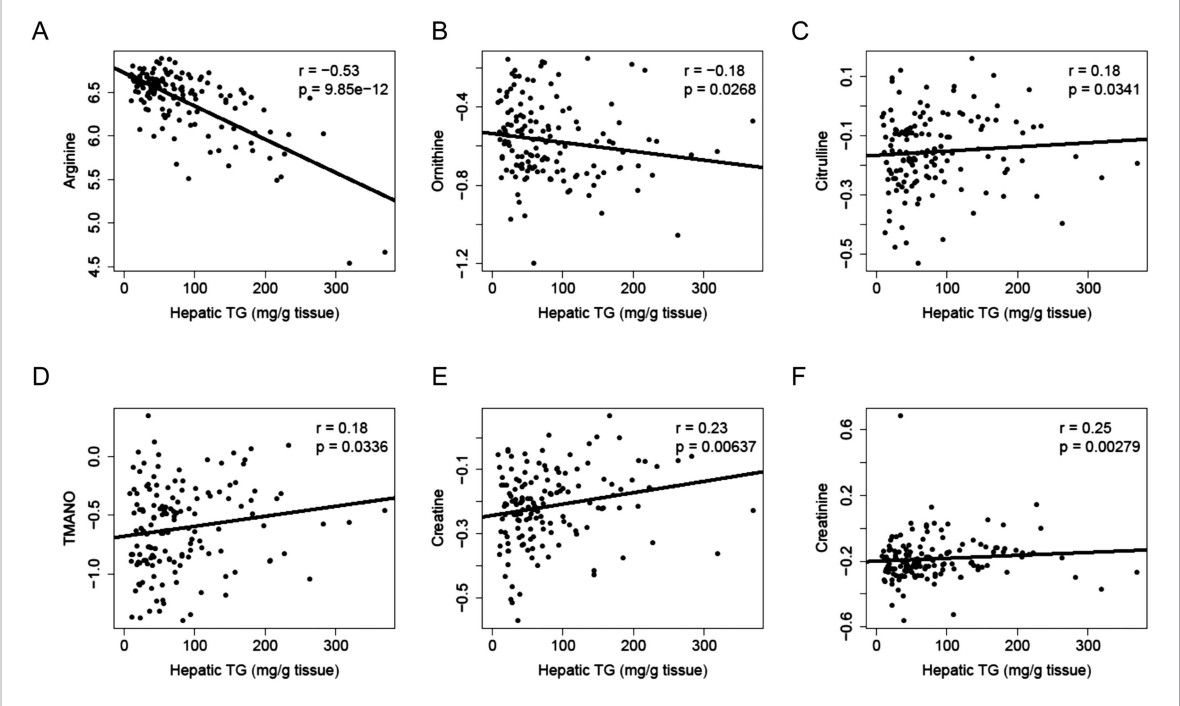

**Figure 10**. Correlation of hepatic TG and polar metabolites in the plasma. Correlation of hepatic TG with plasma levels of arginine (**A**), ornithine (**B**), citrulline (**C**), TMANO (**D**), creatine (**E**), and creatinine (**F**). r, biweight midcorrelation; p, p-value.

Our study has generated a rich resource for future analyses of NAFLD. The primary data are available from the authors and summary tables are posted at http://systems.genetics.ucla.edu.

## Materials and methods

### Animals

The HMDP strains have previously been described in detail (*Bennett et al., 2010*). The strain names and number of mice used in this study are listed in *Table 11*. All mice were obtained from the Jackson Laboratory and bred at University of California, Los Angeles. The experimental design for the diet studies was previously described (*Parks et al., 2013*). Briefly, male mice were maintained on a chow diet (Ralston Purina Company) until 8 weeks of age when they were given a HF/HS diet (Research Diets-D12266B, New Brunswick, NJ) with the following composition: 16.8% kcal protein, 51.4% kcal carbohydrate, 31.8% kcal fat. After 8 weeks on the HF/HS diet, mice were sacrificed after a 4-hr fast. Blood was collected by retro-orbital bleeding under isoflurane anesthesia. Plasma and livers were frozen immediately in liquid nitrogen and stored at −80°C until analysis. All assays and phenotypes were from the same mice. Mice were housed in rooms with a 14-hr light/10-hr dark cycle (light is on between 6 a.m. and 8 p.m.). On the day of experiment, mice were fasted at 6:30 a.m. and sacrificed between 10:30 a.m. and noon. All animal procedures were approved by the Institutional Care and Use Committee (IACUC) at University of California, Los Angeles.

### Hepatic lipid content

Liver lipids were extracted by the method of *Folch et al. (1957)*. About 100 mg of livers was used for lipid extraction and the dried organic extract was dissolved in 1.8% (wt/vol) Triton X-100. The amount of lipids in each extract was determined by colorimetric assay from Sigma (St. Louis, MO) (triglyceride, TC and unesterified cholesterol) and Wako (Richmond, VA) (phospholipids) according to the manufacturer's instructions.

**Table 10**. Correlation between hepatic lipids and gut microbiota

| | TG | TC | UC | PL |
|---|---|---|---|---|
| Family: | | | | |
| *Clostridiaceae* | −0.133 | 0.035 | 0.116 | 0.052 |
| *Erysipelotrichaceae* | 0.138 | 0.005 | −0.057 | −0.016 |
| *Lachnospiraceae* | 0.066 | −0.112 | 0.000 | 0.045 |
| *Mogibacteriaceae* | 0.033 | −0.055 | −0.092 | −0.089 |
| *Peptostreptococcaceae* | 0.006 | −0.019 | −0.003 | 0.096 |
| *Rikenellaceae* | 0.002 | 0.142 | 0.125 | −0.047 |
| *Ruminococcaceae* | 0.138 | 0.176 | 0.169 | 0.030 |
| *S24-7* | −0.092 | 0.003 | −0.111 | −0.166 |
| Genus: | | | | |
| *02d06* | −0.146 | −0.062 | −0.05 | 0.012 |
| *Adlercreutzia* | 0.014 | 0.001 | 0.133 | 0.084 |
| *Akkermansia* | −0.031 | 0.029 | 0.080 | 0.074 |
| *Allobaculum* | −0.083 | −0.205* | −0.178 | −0.009 |
| *Anaeroplasma* | −0.019 | −0.047 | 0.001 | 0.018 |
| *Bifidobacterium* | −0.014 | −0.082 | −0.131 | −0.084 |
| *Blautia* | 0.003 | −0.217* | −0.244† | −0.183 |
| *Clostridium* | −0.108 | −0.063 | −0.044 | −0.008 |
| *Clostridium.1* | 0.081 | 0.044 | −0.023 | 0.015 |
| *Coprobacillus* | −0.008 | −0.075 | −0.138 | −0.130 |
| *Coprococcus* | −0.016 | 0.174 | 0.261† | 0.212* |
| *Dehalobacterium* | 0.086 | 0.070 | 0.066 | 0.070 |
| *Dorea* | 0.078 | 0.070 | 0.056 | 0.081 |
| *Lactobacillus* | 0.005 | 0.021 | −0.012 | −0.073 |
| *Lactococcus* | 0.085 | 0.001 | 0.048 | 0.065 |
| *Oscillospira* | 0.140 | 0.127 | 0.261† | 0.206* |
| *R.Gnavus* | 0.103 | −0.024 | −0.037 | −0.016 |
| *Roseburia* | 0.063 | −0.090 | −0.132 | −0.151 |
| *Ruminococcus* | 0.099 | −0.131 | −0.160 | −0.218* |
| *Sarcina* | −0.109 | 0.035 | 0.067 | 0.014 |
| *SMB53* | −0.012 | −0.069 | −0.029 | 0.052 |
| *Turicibacter* | −0.035 | 0.016 | 0.019 | 0.013 |

TG: triglyceride; TC: total cholesterol; UC: unesterified cholesterol; PL: phospholipids.
*Denotes FDR < 0.05.
†Denotes FDR <0.01.

## ALT activity assay

Plasma ALT activity was assayed using a kinetic colorimetric assay kit from Pointe Scientific, Inc (Canton, MI, USA) according to the manufacturer's protocol. Activity of ALT was determined by the rate of decrease in NADH as measured by the change in absorbance at 340 nm.

## RNA isolation and gene expression analyses

Flash-frozen liver and epididymal adipose samples were weighed and homogenized in Qiazol (Qiagen, Valencia, CA), and RNA was isolated according to the manufacturer's protocol using RNeasy

**Table 11.** Inbred and recombinant inbred strains used in this study

| Inbred strains | | Recombinant inbred strains | | | |
|---|---|---|---|---|---|
| Strain | n | Strain | n | Strain | n |
| 129X1/SvJ | 4 | AXB10/PgnJ | 3 | BXD1/TyJ | 3 |
| A/J | 4 | AXB12/PgnJ | 5 | BXD11/TyJ | 2 |
| AKR/J | 5 | AXB13/PgnJ | 5 | BXD12/TyJ | 5 |
| BALB/cJ | 5 | AXB15/PgnJ | 4 | BXD13/TyJ | 2 |
| BTBR T<+> tf/J | 4 | AXB19/PgnJ | 5 | BXD14/TyJ | 4 |
| BUB/BnJ | 5 | AXB19a/PgJ | 4 | BXD15/TyJ | 5 |
| C3H/HeJ | 4 | AXB19b/PgJ | 5 | BXD16/TyJ | 3 |
| C57BL/6J | 4 | AXB2/PgnJ | 3 | BXD19/TyJ | 4 |
| C57BLKS/J | 4 | AXB5/PgnJ | 4 | BXD20/TyJ | 7 |
| C57L/J | 5 | AXB6/PgnJ | 3 | BXD21/TyJ | 5 |
| C58/J | 4 | AXB8/PgnJ | 5 | BXD24/TyJ | 4 |
| CBA/J | 4 | BXA1/PgnJ | 3 | BXD31/TyJ | 2 |
| CE/J | 4 | BXA11/PgnJ | 4 | BXD32/TyJ | 5 |
| DBA/2J | 4 | BXA12/PgnJ | 4 | BXD34/TyJ | 4 |
| FVB/NJ | 5 | BXA13/PgnJ | 3 | BXD36/TyJ | 4 |
| I/LnJ | 5 | BXA14/PgnJ | 4 | BXD38/TyJ | 4 |
| KK/HlJ | 5 | BXA16/PgnJ | 3 | BXD39/TyJ | 4 |
| LG/J | 4 | BXA2/PgnJ | 5 | BXD40/TyJ | 7 |
| MA/MyJ | 4 | BXA24/PgnJ | 4 | BXD43/RwwJ | 5 |
| NOD/ShiLtJ | 2 | BXA4/PgnJ | 4 | BXD44/RwwJ | 4 |
| NON/ShiLtJ | 5 | BXA7/PgnJ | 6 | BXD45/RwwJ | 5 |
| NZB/BlNJ | 4 | BXA8/PgnJ | 4 | BXD48/RwwJ | 4 |
| NZW/LacJ | 1 | BXH19/TyJ | 6 | BXD49/RwwJ | 4 |
| PL/J | 6 | BXH2/TyJ | 3 | BXD5/TyJ | 5 |
| RIIIS/J | 4 | BXH20/KccJ | 3 | BXD50/RwwJ | 4 |
| SEA/GnJ | 4 | BXH22/KccJ | 4 | BXD51/RwwJ | 4 |
| SJL/J | 5 | BXH4/TyJ | 4 | BXD55/RwwJ | 4 |
| SM/J | 4 | BXH6/TyJ | 5 | BXD56/RwwJ | 6 |
| SWR/J | 4 | BXH8/TyJ | 3 | BXD6/TyJ | 5 |
| | | BXH9/TyJ | 6 | BXD60/RwwJ | 2 |
| | | CXB11/HiAJ | 5 | BXD61/RwwJ | 4 |
| | | CXB12/HiAJ | 5 | BXD62/RwwJ | 6 |
| | | CXB13/HiAJ | 5 | BXD64/RwwJ | 4 |
| | | CXB3/ByJ | 5 | BXD66/RwwJ | 5 |
| | | CXB4/ByJ | 5 | BXD68/RwwJ | 4 |
| | | CXB6/ByJ | 6 | BXD70/RwwJ | 4 |
| | | CXB7/ByJ | 6 | BXD71/RwwJ | 4 |
| | | | | BXD73/RwwJ | 4 |
| | | | | BXD74/RwwJ | 4 |
| | | | | BXD75/RwwJ | 4 |
| | | | | BXD79/RwwJ | 4 |
| | | | | BXD8/TyJ | 2 |

*Table 11. Continued on next page*

columns (Qiagen). Isolated RNA was analyzed for global gene expression using Affymetrix HT_MG430A arrays, and data from the microarray analysis were filtered as described (*Bennett et al., 2010*). Gene enrichment analysis was performed using the Database for Annotation, Visualization and Integrated Discovery (DAVID, v6.7) program (*Huang da et al., 2009*; *Huang da et al., 2009*).

## Genome-wide association analysis

Genotypes for 113 strains of mice (29 classical inbred, 84 recombinant inbred) were obtained from Jackson Laboratories using the Mouse Diversity Array (*Yang et al., 2009*). After removing SNPs that were flagged as poor quality, 459911 SNPs remained. We further filtered these to about 200,000 SNPs by removing SNPs that did not have a minor allele frequency of >5% and a missing genotype rate of <10%. Genome-wide association mapping of the hepatic TG content was performed using FaST-LMM (Factored Spectrally Transformed Linear Mixed Models), which uses a linear mixed model to correct for population structure (*Listgarten et al., 2012*). Cut-off values for genome-wide significance were determined by computing the FDR estimated by the q values (*Storey and& Tibshirani, 2003*).

## Generation of adenovirus and in vivo transduction

Recombinant adenovirus was generated using the AdEasy system as previously described (*Bennett et al., 2013*). Briefly, a shuttle vector containing the full-length mouse *Gde1* cDNA and a C-terminal *c-myc* tag or bacterial LacZ sequence was cotransformed with the adenoviral backbone plasmid pAdEasy-1 for homologous recombination in *Escherichia coli* BJ5183 cells. Positive recombinants were linearized and transfected into 293 cells for virus packaging and propagation. Adenoviruses were purified by CsCl banding and stored at −80°C until use. Adenovirus expressing shRNA for *Gde1* (5′-CCGGGACATCGAGTTTACTTCTGATCTCGA-GATCAGAAGTAAACTCGATGTCTTTTTG-3′) driven by a U6 promoter was generated in a similar fashion. For adenoviral infection, 8-week-old chow-fed male C57BL/6 mice (8 per group) were injected with adenoviral construct (1 × 10$^9$ pfu diluted in 0.2 ml saline, *i.v.*) via the tail vein. After injection, the mice were switched to a HF/HS diet. The control group consisted of mice injected with adenoviral construct expressing the LacZ gene in the overexpression studies

*Table 11. Continued*

| Inbred strains | | Recombinant inbred strains | | | |
|---|---|---|---|---|---|
| **Strain** | **n** | **Strain** | **n** | **Strain** | **n** |
| | | | | BXD84/RwwJ | 4 |
| | | | | BXD85/RwwJ | 4 |
| | | | | BXD86/RwwJ | 5 |
| | | | | BXD87/RwwJ | 3 |
| | | | | BXD9/TyJ | 4 |

or empty virus in shRNA knockdown experiments. Mice were sacrificed 7 days post injection after a 4-hr fast. Expression of *Gde1* in the liver was assessed by qPCR and Western blotting. Data from mice showing the absence of over-expression or knockdown were excluded from the analysis.

## Quantitation of plasma metabolites by metabolomics

Metabolic profiling by LC-MS of amino acids, biogenic amines, and other polar metabolites in plasma was performed as previously described (*Wang et al., 2011*; *Roberts et al., 2012*). Metabolite concentrations were determined using the standard addition method (*Ito and Tsukada, 2002*).

## Gut microbiota analysis

Microbial DNA was isolated from the cecum using the PowerSoil DNA Isolation Kit according to the manufacturer's instructions (MO BIO Laboratories, Carlsbad, CA). Region-specific primers including the Illumina flowcell adapter sequences were used for amplifying the V4 region of the 16S rRNA gene. The reverse amplicon primer contains a 12-base barcode sequence that allows sample pooling for sequencing. The barcoded primers and sample preparation were performed as previously described (*Hamady et al., 2008*; *Costello et al., 2009*; *Caporaso et al., 2012*). Each sample was amplified in triplicate, combined and cleaned using the PCR clean-up kit (MO BIO Laboratories, Carlsbad, CA). Cleaned and quantified amplicons were sequenced with the Illumina MiSeq machine at the GenoSeq Core Facility at the University of California, Los Angeles using 500-cycle PE kit. The MiSeq run contained a control library, which was made from phiX174 as described (*Caporaso et al., 2012*). The raw data from the MiSeq run were first processed through a quality filter using established guidelines (*Bokulich et al., 2013*). The remaining sequences were analyzed using the open source software package Quantitative Insights Into Microbial Ecology (QIIME) version 1.7.0 (*Caporaso et al., 2010*; *Kuczynski et al., 2011*). Demultiplexed sequences from all of the samples were clustered into operational taxonomic units (OTUs) based on their sequence similarity (97% identity) using a reference based OTU picking protocol in QIIME. The taxonomic composition was assigned to the representative sequence of each OTU using Ribosomal Database Project (RDP) Classifier 2.0.1 (greengenes 13-08) (*Wang et al., 2007*). The relative abundance of bacteria at each taxonomic level (e.g., phylum, class, order, family, and genus) was computed for each mouse. Biweight midcorrelation (bicor) values between microbiota and measured traits were determined using R statistic program.

## Acknowledgements

We thank Hannah Qi, Zhiqiang Zhou, Judy Wu, and Tieyan Han for their expert assistance with mouse experiments.

## Additional information

### Funding

| Funder | Grant reference | Author |
|---|---|---|
| National Heart, Lung, and Blood Institute (NHBLI) | HL028481 | Aldons J Lusis |
| European Commission (EC) | FP7 - 330381 | Elin Org |
| Norges Forskningsråd | 240405/F20 | Frode Norheim |
| National Institute of Diabetes and Digestive and Kidney Diseases (NIDDK) | DK094311 | Aldons J Lusis |

| Funder | Grant reference | Author |
|---|---|---|
| National Institutes of Health (NIH) | T32-HD07228 | Brian W Parks |

The funders had no role in study design, data collection and interpretation, or the decision to submit the work for publication.

## Author contributions

STH, BWP, Conception and design, Acquisition of data, Analysis and interpretation of data, Drafting or revising the article; EO, FN, NC, CP, LWC, SC, DLD, NP, IN, REG, TK, PSG, Acquisition of data, Analysis and interpretation of data, Drafting or revising the article; AJL, Conception and design, Analysis and interpretation of data, Drafting or revising the article

## Ethics

Animal experimentation: This study was performed in strict accordance with the recommendations in the Guide for the Care and Use of Laboratory Animals of the National Institutes of Health. All of the animals were handled according to approved institutional animal care and use committee (IACUC) protocols (#92-169) of the University of California at Los Angeles.

# Additional files

## Supplementary files

• Supplementary file 1. Strain average of hepatic lipids.

• Supplementary file 2. Correlation of hepatic TG and plasma metabolites.

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
