## [Decision Letter]

Thank you for sending your work entitled “The Genetic Architecture of NAFLD among Inbred Strains of Mice” for consideration at *eLife*. Your article has been favorably evaluated by Diethard Tautz (Senior editor) and three reviewers, one of whom is a member of our Board of Reviewing Editors.

The Reviewing editor and the other reviewers discussed their comments before we reached this decision, and the Reviewing editor has assembled the following comments to help you prepare a revised submission.

The manuscript from Hui et al. reports on a systems genetics approach to identify genes that influence hepatic lipid content in inbred strains of mice fed a high fat, high sucrose diet for eight weeks. They systematically collect and analyze an impressive array of data, including physiologic/anatomic parameters (e.g., liver TGs, blood glucose), SNP genotype, liver and adipose tissue gene expression, metabolites and gut microbe 16S RNA sequence. The physiologic/anatomic, SNP and gene expression data is integrated to identify candidate genes that affect hepatic lipid content, including *Gde1*. They demonstrate that overexpression of *Gde1* in liver results in increased hepatic triglyceride and alterations in hepatic gene expression that support a causal role in regulation of liver fat content. All three reviewers were enthusiastic about the main findings and their significance. One reviewer commented: 'This is an excellent genetic survey of hepatic steatosis in a collection of recombinant inbred strains that have been used as a gene mapping resource'. A second reviewer commented: 'An important body of work and an excellent manuscript'. The following major concerns need to be addressed.

1) The main biological finding to emerge from these studies is the identification and validation of *Gde1* as a novel gene that in some way controls hepatic TG content. The synthesis of genome-wide association and eQTL analysis makes a strong case, the validation studies are not yet convincing. In Figure 10, the expression levels of *Gde1* in liver lie between 8 and 10 on the X axis, which is undefined. Assuming a log2 scale, the variation in expression of *Gde1* is less than 4-fold across the 100+ strains. The Western blot of *Gde1* in Figure 11A shows no detectable *Gde1* in the Adeno-control livers and high levels in the Adeno-Gde1 livers, implying that the differences in expression are much greater and that *Gde1* is being expressed at a supra-physiologic level. The functional validation of *Gde1* therefore needs to be further validated, ideally by a loss of function experiment, e.g., using the same adenoviral system to deliver shRNAs, or use of ASOs, which effectively target the liver. In addition, measurements of glycerol-3-phosphate could be performed in these studies to investigate the proposed mechanism by which *Gde1* influences TG content.

2) The authors state that they demonstrate vast genetic variation in hepatic TG accumulation in response to a HF/HS diet. This is not the case, however, because the hepatic TG contents (or other values) are not reported under control conditions. This data may be available, but unless results are reported as changes from a control diet, the values observed in the HF/HS diet cannot be considered to be a 'response'. While this does not detract from the results, the point should be clarified in the revised manuscript.

3) The SNPs associated with *Gde1* eQTLs in liver are different than in adipose tissue, and *Gde1* expression in liver is positively correlated with liver TG, while *Gde1* expression in adipose tissue is negative correlated. Although the SNPs in question are unlikely to be causative, do the corresponding regions of the genome exhibit potential regulatory elements in liver and adipose tissue? Given that eQTLs for *Gde1* in adipose are more significant than in liver, can the authors comment further on the extent to which candidate genes identified in these studies might regulate TG content in liver through actions in adipose tissue?

Minor comments:

1) Introduction, second paragraph. The implication of these statements seems to be that the 2-hit hypothesis is insufficient to explain NAFLD development in all cases, yet in the same paragraph, the “second hit” is referenced again. It is not entirely clear how these statements fit together. An additional sentence explaining the interpretation of the DGAT2 results in light of the 2-hit hypothesis would help to clarify this section.

2) Introduction, last paragraph. The term “heritable” implies transmission from parents to offspring, yet none of the experiments included multiple generations.

3) In the subsection headed “Animals”. What was the specific method of blood collection?

4) In the subsection headed “Measurement of hepatic lipid content”. It's too bad that the study was not conducted for a longer period so that susceptibility to NASH could be assessed NAFLD is a common but benign condition. It is important primarily because a substantial number of cases transition to NASH, but who transitions is currently unpredictable.

5) In the subsection headed “Alanine aminotransferase (ALT) activity assay”. ALT as a measure of liver damage—a standard, but sometimes unreliable, marker. Histology should be presented too.

6) In the subsection headed “Large genetic variation in hepatic TG accumulation in response to a HF/HS diet” and elsewhere. Significant trait correlations are reported as 'strong'. But although these are often statistically significant, the fraction of phenotypic variation is quite modest. For example, liver TG and phospholipids, r = -0.26, P<10-6, but r2 is <7%. Statements about strong correlations need be tempered in language and interpretation.

7) In the subsection headed “Large genetic variation in hepatic TG accumulation in response to a HF/HS diet”. It would be helpful to list the number of mice analyzed per cohort (or at least the minimum number used).

8) In the subsection headed “Large genetic variation in hepatic TG accumulation in response to a HF/HS diet”. It is not clear if the reported p-values were adjusted for multiple hypothesis testing, which needs to be done.

9) In the subsection headed “Large genetic variation in hepatic TG accumulation in response to a HF/HS diet”. The lack of large variation in cholesterol and phospholipid levels among strains despite correlation with hepatic TG content (which does show large variation) makes it seem like the correlations are not very strong.

10) In the subsection headed “Large genetic variation in hepatic TG accumulation in response to a HF/HS diet”. A lack of enrichment of three specific immune cell markers is used as evidence for a lack of hepatic infiltration in steatotic livers, but it is quite possible that other immune cell types are present. Indeed, the subsection headed “Transcriptomic analysis of the liver and adipose tissue” reports that several genes expressed by inflammatory cells are increased in steatotic livers. A more thorough assessment of immune cell marker gene expression should be included.

11) In the subsection headed “Transcriptomic analysis of the liver and adipose tissue”. Are any of the differentially expressed genes closely linked; could local epigenetic changes lead to local changes in gene expression, raising the possibility that some changes are 'noise'?

12) In the same subsection. Do strains showing contrasting mRNA levels share haplotypes, e.g. high strains share one haplotype at a particular locus and low strains another haplotype? And are these allelic differences associated with underlying DNA sequence variants?

13) Still in the same subsection. Is expression variation at a given locus bimodal (allelic effects?) or uniform (perhaps a quantitative trait with multilocus control)?

14) In the subsection headed “Transcriptomic analysis of the liver and adipose tissue”. In addition to the analysis provided, it may be useful to perform two independent pathway enrichment analyses on genes that are positively or negatively correlated with hepatic TG, respectively.

15) At the end of the first paragraph of the subsection “Integration of transcriptomic and association mapping data”. It is unclear what is meant by the term “conservative” here. Was an analysis performed to predict the functional consequences of each missense variant?

16) In the subsection entitled “Adenovirus-mediated expression of *Gde1* increases liver TG accumulation”. What is the justification for using a 7 day HF/HS diet feeding period when the initial experiments used an 8 week period?

17) In the subsection “Metabolomic profiling of plasma metabolites”. It would be interesting to determine correlations between gene expression and levels of specific metabolites or microbes, as was done for the hepatic phenotypes.

18) In the subsection headed “Association of gut microbiota with hepatic lipid accumulation”. Were there any correlations with overall diversity of microbial communities?

19) Discussion. More discussion should be included regarding the potential implications of the correlations that were identified. How do these findings reflect the mechanisms underlying NAFLD development and progression, as well as their differences between strains?

20) Throughout. Were all of the assays conducted contemporaneously in the same mice? Or were these different cohorts, different times? Were results of some assays reported previously? The authors have a long-term commitment to this important paradigm. They just need to be clear about what was done when, with which cohorts.

---

## [Author Response]

*1) The main biological finding to emerge from these studies is the identification and validation of* Gde1 *as a novel gene that in some way controls hepatic TG content. The synthesis of genome-wide association and eQTL analysis makes a strong case, the validation studies are not yet convincing. In*
Figure 10*, the expression levels of* Gde1 *in liver lie between 8 and 10 on the X axis, which is undefined. Assuming a log2 scale, the variation in expression of* Gde1 *is less than 4-fold across the 100+ strains. The Western blot of* Gde1 *in Figure 11A shows no detectable* Gde1 *in the Adeno-control livers and high levels in the Adeno-*Gde1 *livers, implying that the differences in expression are much greater and that* Gde1 *is being expressed at a supra-physiologic level. The functional validation of* Gde1 *therefore needs to be further validated, ideally by a loss of function experiment, e.g., using the same adenoviral system to deliver shRNAs, or use of ASOs, which effectively target the liver. In addition, measurements of glycerol-3-phosphate could be performed in these studies to investigate the proposed mechanism by which* Gde1 *influences TG content*.

We have performed additional studies using shRNA knockdown in vivo to validate *Gde1* as the casual gene for steatosis (Figure 9) Mice receiving adenovirus expressing shRNA for *Gde1* showed a 75% reduction in *Gde1* expression. Compared to control mice, *Gde1* knocked down mice exhibited ∼30% decreased in hepatic TG contents. These data are complimentary to the overexpression studies (Figure 8), showing a positive correlation of hepatic TG levels with *Gde1* expression.

Since glycerol-3-phosphate is an intermediate in the crossroad of several metabolic pathways, it is unlikely that its steady state level will be elevated due to increased *Gde1* activity. Detailed analysis of the metabolic flux (e.g. using isoprotomer labeling) will be required to delineate the mechanism, and we feel that this is beyond the scope of this paper, which focuses on the genetic aspects of NALFD.

*2) The authors state that they demonstrate vast genetic variation in hepatic TG accumulation in response to a HF/HS diet. This is not the case, however, because the hepatic TG contents (or other values) are not reported under control conditions. This data may be available, but unless results are reported as changes from a control diet, the values observed in the HF/HS diet cannot be considered to be a 'response'. While this does not detract from the results, the point should be clarified in the revised manuscript*.

We have data from a small number of strains fed the control diet, and the TG levels are uniformly low (<20 mg/g liver). Therefore, we assumed that final TG reflected the response to the diet. We have now clarified the point as requested.

*3) The SNPs associated with* Gde1 *eQTLs in liver are different than in adipose tissue, and* Gde1 *expression in liver is positively correlated with liver TG, while* Gde1 *expression in adipose tissue is negative correlated. Although the SNPs in question are unlikely to be causative, do the corresponding regions of the genome exhibit potential regulatory elements in liver and adipose tissue? Given that eQTLs for* Gde1 *in adipose are more significant than in liver, can the authors comment further on the extent to which candidate genes identified in these studies might regulate TG content in liver through actions in adipose tissue?*

The opposite pattern of regulation of *Gde1* implicates tissue-specific regulatory elements. Mouse ENCODE data showed that there are differences in DNA hypersensitive sites in the gene region, suggesting that there may be differences in DNA accessibility and transcription factor binding in the liver and adipose tissue.

The negative correlation of *Gde1* and hepatic TG may be due to a different role of glycerol-3-phosphate in the adipose tissue. Increased glycerol-3-phosphate production (when *Gde1* expression is high) in the adipose tissue could promote fatty acid re-esterification, leading to a decrease in fatty acid release to the circulation. This reduction in fatty acid supply to the liver may result in diminished TG synthesis. Discussion of these points is now included.

Minor comments:

*1) Introduction, second paragraph. The implication of these statements seems to be that the 2-hit hypothesis is insufficient to explain NAFLD development in all cases, yet in the same paragraph, the* “*second hit*” *is referenced again. It is not entirely clear how these statements fit together. An additional sentence explaining the interpretation of the DGAT2 results in light of the 2-hit hypothesis would help to clarify this section*.

We have further elaborated the results from the DGAT2 knockdown studies. Their paradoxical findings suggest that TG accumulation per se is not the “first hit” but rather that it is the underlying inability to compensate for increased FA flux which makes the liver prone to subsequent oxidative damage.

*2) Introduction, last paragraph. The term* “*heritable*” *implies transmission from parents to offspring, yet none of the experiments included multiple generations.*

We assume that since the environmental is the same but the phenotypes differ, genetics most account for the differences. Nevertheless, we have now substituted “dependent on genetic background” for “heritable”.

3) In the subsection headed “Animals”. What was the specific method of blood collection?

Blood was collected by retro-orbital bleeding under isoflurane anesthesia. This information has been included in Methods.

*4) In the subsection headed “Measurement of hepatic lipid content”. It's too bad that the study was not conducted for a longer period so that susceptibility to NASH could be assessed NAFLD is a common but benign condition. It is important primarily because a substantial number of cases transition to NASH, but who transitions is currently unpredictable*.

We agree with the reviewer’s comment. A longer HF/HS feeding regimen will be considered in future studies.

*5) In the subsection headed “Alanine aminotransferase (ALT) activity assay”. ALT as a measure of liver damage—a standard, but sometimes unreliable, marker. Histology should be presented too*.

ALT is an established biochemical and clinical marker for liver damage as this enzyme is released into the circulation when the integrity of the cell membrane of hepatocytes is compromised. However, in advanced liver disease with cirrhosis or necrosis, plasma ALT could be low, giving a false indicator of liver damage. However, in our studies, mice were only on HF/HS diet for 8 weeks and did not show signs of advanced NAFLD. It is unlikely that this confounding factor would significantly affect our conclusion (please see the subsection headed “Large genetic variation in hepatic TG accumulation in mice fed with HF/HS diet”).

Liver samples were not fixed at the time of collection. We examined frozen sections of the liver from selected strains having high and low ALT levels. Although the preservation of tissue structure was less than ideal, no apparent difference in general hepatocyte architecture was observed between the two groups. In addition, evidence of advanced NAFLD (fibrosis and necrosis) was not observed in all samples examined.

*6) In the subsection headed “Large genetic variation in hepatic TG accumulation in response to a HF/HS diet” and elsewhere. Significant trait correlations are reported as 'strong'. But although these are often statistically significant, the fraction of phenotypic variation is quite modest. For example, liver TG and phospholipids, r = -0.26, P<10-6, but r2 is <7%. Statements about strong correlations need be tempered in language and interpretation*.

We agree. Wordings of “strong” have been deleted or modified accordingly throughout the manuscript.

*7) In the subsection headed “Large genetic variation in hepatic TG accumulation in response to a HF/HS diet”. It would be helpful to list the number of mice analyzed per cohort (or at least the minimum number used)*.

Information on the strain names and number of mice used in this study is now listed in Table 11.

*8) In the subsection headed “Large genetic variation in hepatic TG accumulation in response to a HF/HS diet”. It is not clear if the reported p-values were adjusted for multiple hypothesis testing, which needs to be done*.

These p-values were not adjusted for multiple testing. These particular correlation analyses were performed based on knowledge of potential association between NAFLD and those clinical traits (e.g. insulin resistance, plasma lipids and adiposity). Nevertheless, the correlations remained significant after Bonferroni correction. This is now discussed at the end of the subsection “Hepatic TG accumulation is highly correlated with plasma cholesterol, insulin resistance and adiposity”.

*9) In the subsection headed “Large genetic variation in hepatic TG accumulation in response to a HF/HS diet”. The lack of large variation in cholesterol and phospholipid levels among strains despite correlation with hepatic TG content (which does show large variation) makes it seem like the correlations are not very strong*.

The correlation between hepatic TG and total cholesterol was 0.35, whereas the correlation with phospholipids was -0.26. These correlations were modest and we have modified the text to reflect this.

*10) In the subsection headed “Large genetic variation in hepatic TG accumulation in response to a HF/HS diet”. A lack of enrichment of three specific immune cell markers is used as evidence for a lack of hepatic infiltration in steatotic livers, but it is quite possible that other immune cell types are present. Indeed, the subsection headed “Transcriptomic analysis of the liver and adipose tissue” reports that several genes expressed by inflammatory cells are increased in steatotic livers. A more thorough assessment of immune cell marker gene expression should be included*.

In addition to Cd45 and Cd 68 reported in the manuscript, other markers for immune cells (T cells, B cells and leukocytes) did not show significant correlation in hepatic TG contents. These include:

*Cd28, Csf2, Cd4, Ccr5, Gata3 Cxcr4* (T cells)

*Pax5, Cd70, Cd79b* (B cells)

*Cd33, Cd52, Cd53, Cd44, Prg2* (Leukocytes)

This information has been included in the revised manuscript.

11) In the subsection headed “Transcriptomic analysis of the liver and adipose tissue”. Are any of the differentially expressed genes closely linked; could local epigenetic changes lead to local changes in gene expression, raising the possibility that some changes are 'noise'?

We looked at the chromosome locations of these genes and found that none of the genes are in close proximity (<1.5 Mb) to each other on the same chromosome.

12) In the same subsection. Do strains showing contrasting mRNA levels share haplotypes, e.g. high strains share one haplotype at a particular locus and low strains another haplotype? And are these allelic differences associated with underlying DNA sequence variants?

The eQTL mapping that we performed addresses the question. Most of the large difference in gene expression are due to local differences and are termed cis-eQTL. We have addressed this in more detail in Orozo et al. (Cell. 2012 151(3):658-70; please see the subsection entitled “Integration of transcriptomic and association mapping data”).

13) Still in the same subsection. Is expression variation at a given locus bimodal (allelic effects?) or uniform (perhaps a quantitative trait with multilocus control)?

The expression variation of the three candidate genes (*Gde1*, *Knop1* and *Coq7*) in the chromosome 7 locus showed a continuous spectrum across the strains. The expression pattern indicates that the expression variations are not bimodal (first paragraph of the subsection headed “Integration of transcriptomic and association mapping data”).

Author response image 1.Variation in candidate gene expression among strains.The expression level of *Gde1* (A), *Knop1* (B) and *Coq7* (C) among HMDP strains are shown. Results are presented as mean ± SD in log2 scale.**DOI:**
http://dx.doi.org/10.7554/eLife.05607.028

*14) In the subsection headed “Transcriptomic analysis of the liver and adipose tissue”. In addition to the analysis provided, it may be useful to perform two independent pathway enrichment analyses on genes that are positively or negatively correlated with hepatic TG, respectively*.

We performed the separate pathway enrichment analyses on positively or negatively correlated genes respectively but did not find any additional enriched pathways that differ from the combined results presented in Table 3 and Table 4.

*15) At the end of the first paragraph of the subsection “Integration of transcriptomic and association mapping data”. It is unclear what is meant by the term* “*conservative*” *here. Was an analysis performed to predict the functional consequences of each missense variant?*

The term “conservative” has been changed to “neutral” to better describe the amino acid change to be non-deleterious. The effect of amino acid substitution on the biological function of the protein was assessed by PROVEAN (Protein Variation Effect Analyzer) prediction tool.

*16) In the subsection entitled “Adenovirus-mediated expression of* Gde1 *increases liver TG accumulation”. What is the justification for using a 7 day HF/HS diet feeding period when the initial experiments used an 8 week period?*

This regimen was chosen because gene expression by adenoviral transduction is only sustained for a short period of time (a few weeks or less). Preliminary studies in mice showed that HF/HS diet induced a 3-fold increase in hepatic TG accumulation in 1 week. A 2-week feeding gave similar results to 1-week feeding. Therefore, the shorter time regimen (1 week) was chosen for this particular study. This is now discussed in the subsection headed “Validation of *Gde1* as a causal gene for hepatic steatosis at the chromosome 7 locus using adenoviral overexpression of *Gde1* and shRNA knockdown”.

*17) In the subsection “Metabolomic profiling of plasma metabolites”. It would be interesting to determine correlations between gene expression and levels of specific metabolites or microbes, as was done for the hepatic phenotypes*.

This is a good idea and we have considered it. However, because of the problem of multiple comparisons (many genes and microbes), no significant associations were observed.

18) In the subsection headed “Association of gut microbiota with hepatic lipid accumulation”. Were there any correlations with overall diversity of microbial communities?

There is no significant correlation between hepatic TG levels and overall microbial diversity (as determined by Shannon diversity index).

19) Discussion. More discussion should be included regarding the potential implications of the correlations that were identified. How do these findings reflect the mechanisms underlying NAFLD development and progression, as well as their differences between strains?

We have added discussion on the role of mitochondria in NAFLD pathogenesis.

In addition, the connection between arginine and gut microbiota linking to NAFLD has also been added.

*20) Throughout. Were all of the assays conducted contemporaneously in the same mice? Or were these different cohorts, different times? Were results of some assays reported previously? The authors have a long-term commitment to this important paradigm. They just need to be clear about what was done when, with which cohorts*.

All assays and phenotypes were from the same mice. Liver lipid measurements were done in a subset of a larger cohort of mice used in our obesity study described previously in Parks et al. (Cell Metab. 2013 8;17(1):141-52).